# Characterizing meiotic chromosomes' structure and pairing using a designer sequence optimized for Hi-C

Héloïse Muller[1,2,3,†,‡], Vittore F Scolari[1,2,3,‡], Nicolas Agier[4], Aurèle Piazza[1,2,3], Agnès Thierry[1,2,3], Guillaume Mercy[1,2,3] (iD), Stéphane Descorps-Declère[3], Luciana Lazar-Stefanita[1,2,3] (iD), Olivier Espeli[5], Bertrand Llorente[6], Gilles Fischer[4], Julien Mozziconacci[7,*] (iD) & Romain Koszul[1,2,3,**] (iD)

## Abstract

In chromosome conformation capture experiments (Hi-C), the accuracy with which contacts are detected varies due to the uneven distribution of restriction sites along genomes. In addition, repeated sequences or homologous regions remain indistinguishable because of the ambiguities they introduce during the alignment of the sequencing reads. We addressed both limitations by designing and engineering 144 kb of a yeast chromosome with regularly spaced restriction sites (Syn-HiC design). In the Syn-HiC region, Hi-C signal-to-noise ratio is enhanced and can be used to measure the shape of an unbiased distribution of contact frequencies, allowing to propose a robust definition of a Hi-C experiment resolution. The redesigned region is also distinguishable from its native homologous counterpart in an otherwise isogenic diploid strain. As a proof of principle, we tracked homologous chromosomes during meiotic prophase in synchronized and pachytene-arrested cells and captured important features of their spatial reorganization, such as chromatin restructuration into arrays of Rec8-delimited loops, centromere declustering, individualization, and pairing. Overall, we illustrate the promises held by redesigning genomic regions to explore complex biological questions.

**Keywords** chromatin loop; cohesin; meiosis; Rec8; synthetic chromosome
**Subject Categories** Chromatin, Epigenetics, Genomics & Functional Genomics; Genome-Scale & Integrative Biology; Methods & Resources
**Mol Syst Biol. (2018) 14:** e8293

## Introduction

Genomic derivatives of the capture of chromosome conformation assay (3C, Hi-C, Capture-C) (Dekker *et al*, 2002; Lieberman-Aiden *et al*, 2009; Hughes *et al*, 2014) are widely applied to decipher the average intra- and inter-chromosomal organization of eukaryotes and prokaryotes (Dekker & Mirny, 2016). Formaldehyde cross-linking followed by segmentation of the genome by a restriction enzyme (RE) is the first step of the experimental protocol. The basic unit of "C" experiments therefore consists of restriction fragments (RFs) that are subsequently re-ligated and captured to identify long-range contacts. The best resolution that can be obtained is directly imposed by the positions of the RE sites along the genome. Both 6-cutter and 4-cutter REs have been used (Sexton *et al*, 2012; Le *et al*, 2013; Marie-Nelly *et al*, 2014; Rao *et al*, 2014), the latter with the expectation that the resolution increases with the number of sites. However, this approach suffers from two major caveats. First, restriction sites (RSs) are not regularly spaced along genomes. The distribution of RF lengths follows a geometric distribution, with important variations along the genome that depend on the local GC content and the specific sequence recognized by the RE. Given that the likelihood for a RF to be cross-linked by formaldehyde during the first step of the procedure depends on its length (Cournac *et al*, 2012), the probability to detect a given fragment in any 3C experiment will be strongly affected by this parameter (Fig 1A). Computational procedures have been developed to correct the signal (Yaffe & Tanay, 2011; Cournac *et al*, 2012; Imakaev *et al*, 2012). Typically, normalization involves filtering out fragments with unusually low or high signal and aggregating the contact data over several consecutive fragments in longer bins of fixed genomic length, at the expense of actual resolution (Lajoie *et al*, 2015). As a consequence, the definition of Hi-C resolution has remained somehow empiric, because of the lack of a control sequence where RF size biases

1 Department Genomes and Genetics, Groupe Régulation Spatiale des Génomes, Institut Pasteur, Paris, France
2 CNRS, UMR 3525, Paris, France
3 Center of Bioinformatics, Biostatistics and Integrative Biology (C3BI), Institut Pasteur, Paris, France
4 Laboratory of Computational and Quantitative Biology, CNRS, Institut de Biologie Paris-Seine, Sorbonne Université, Paris, France
5 Centre Interdisciplinaire de Recherche en Biologie, Collège de France, UMR-CNRS 7241, INSERM U1050, Paris, France
6 Cancer Research Center of Marseille, CNRS UMR7258, Inserm U1068, Institut Paoli-Calmettes, Aix-Marseille Université UM105, Marseille, France
7 Theoretical Physics for Condensed Matter Lab, CNRS UMR 7600, Sorbonne Universités, UPMC University Paris 06, Paris, France
*Corresponding author. Tel: +33 1 44 27 45 40; E-mail: mozziconacci@lptmc.jussieu.fr
**Corresponding author. Tel: +33 1 40 61 33 25; E-mail: romain.koszul@pasteur.fr
‡These authors contributed equally to this work
†Present address: UMR3664 Dynamique du Noyau, Institut Curie, Paris, France

**Figure 1. Syn-HiC design and assembly.**

A Number of contacts made by RFs as a function of their size (HindIII (red) or DpnII (blue) in the native sequence; left panel: log-lin scale; right panel: log-log scale).

B Illustration of the design principles of the Syn-HiC sequence for the DpnII and HindIII RSs. Black arrow: chromosome. Gray rectangles: genetic elements. Blue and red vertical lines represent the RSs' positions for the enzymes DpnII and HindIII, respectively. Top panel: restriction pattern of a (hypothetical) native sequence. Bottom panel: restriction pattern after Syn-HiC design, with the RSs defining regularly spaced intervals.

C Distribution of the DpnII (left) and HindIII (right) RF sizes in the Syn-HiC 150-kb redesigned sequence and for the full chromosome IV (blue and red, respectively).

D Raw DpnII contact map of the Hi-C experiment performed on G1 daughter cells synchronized through elutriation. Dashed lines: borders of the redesigned region. Plain black lines: borders of the contact map analyzed in Fig 2.

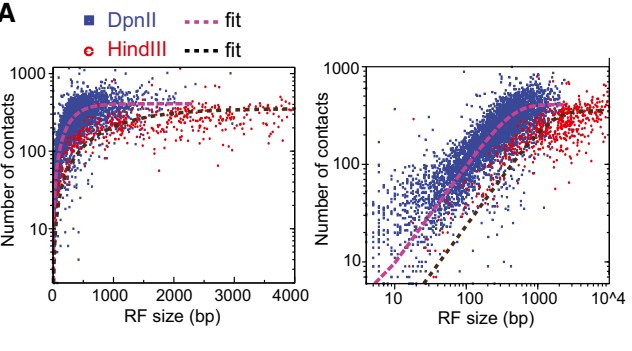

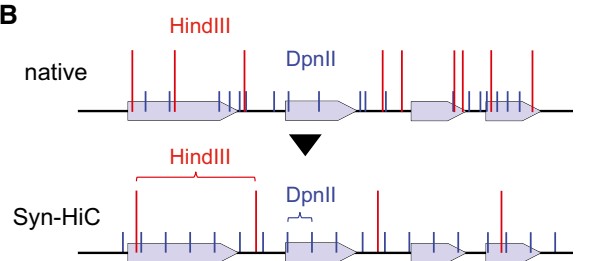

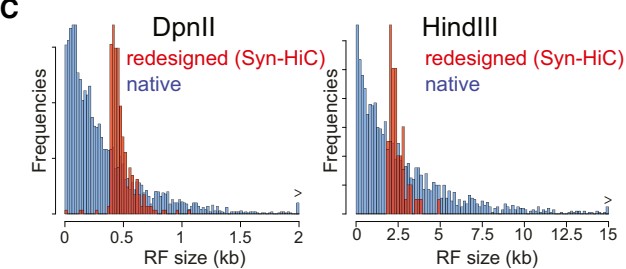

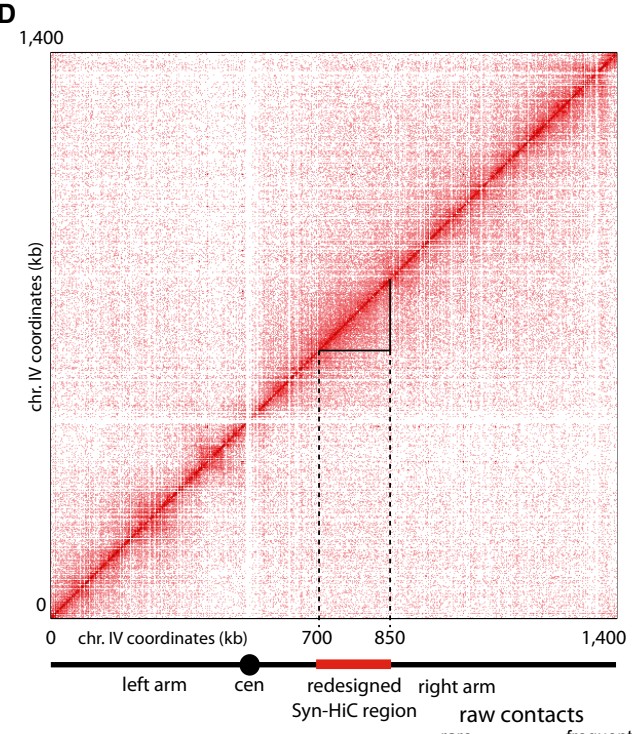

would be alleviated. The second limitation is common to all genomic approaches and reflects the fact that identical sequences cannot be tracked because the sequencing reads cannot be mapped unambiguously along the genome, abolishing the possibility to track homologous chromosomes in isogenic backgrounds.

Approaches using modified restriction patterns have been used to increase/improve the resolution of 3C-based approaches, such as DNAse-HiC and Micro-C (Hsieh *et al*, 2015; Ramani *et al*, 2016). DNAse-HiC captures contacts between open chromatin sites sensitive to DNAse. These sites are found approximately every 3 kb along the yeast genome (Ma *et al*, 2015), and therefore, the resolution reachable through DNAse-HiC remains limited. Micro-C, on the other hand, exploits micrococcal nuclease (MNase) to digest DNA rather than a restriction enzyme (Hsieh *et al*, 2015, 2016). This approach generates non-specific cuts in between nucleosomes (every ~160 bp), resulting in a relatively regular restriction pattern, with heterogeneities resulting from the distribution of nucleosome-free regions. Short-range chromosome contacts captured by Micro-C identified large chromosomal domains within the yeast genome separated by highly expressed genes. While providing a high resolution of intra-chromosomal contacts, this approach nevertheless suffers from the same limitations as classical Hi-C to track homologs and repeated regions.

One consequence of these limitations has been the absence of in-depth studies of meiotic prophase through Hi-C approach, despite the pioneering 3C work (Dekker *et al*, 2002; below). Meiosis is the cell division where a diploid cell gives rise to four haploid gametes through two rounds of chromosome segregation with no intervening replication. The prophase of the first division, where the homologous paternal and maternal chromosomes segregate, comprises a series of regulated events involving homologs condensation, recognition, pairing, and synapsis by the synaptonemal complex (SC) all along their length. Replicated meiotic chromosomes restructure as arrays of chromatin loops anchored at a semi-rigid axis composed of various proteins including cohesins. This restructuration has been observed by cytological approaches in a wide variety of organisms including *S. cerevisiae* (Møens & Pearlman, 1988; Zickler & Kleckner, 1999). The periodic enrichment into discrete domains along *S. cerevisiae* chromosomes of axis-structuring component proteins (Hop1 and Red1) also supports the existence of arrays of chromatin loops anchored

at the axis (Zickler & Kleckner, 1999, 2016; Blat *et al*, 2002). Cytological assays led to an estimated loop size of 20 kb in *S. cerevisiae*, although the limited resolution of the technique impaired a precise characterization of loop features in yeast (Møens & Pearlman, 1988). The meiosis-specific cohesion subunit Rec8 (Klein *et al*, 1999) also follows the periodic enrichment of axis proteins and contributes to the establishment of cohesion between sister chromatids (Panizza *et al*, 2011).

During the meiotic program shared by budding yeasts and mammals (i.e., the succession of events mediated by evolutionarily conserved molecular complexes), condensed chromosomes provide a highly organized architecture for the formation and resolution of DNA double-strand breaks (DSBs) to ultimately make crossovers (COs), essential features for accurate chromosome segregation at meiosis I. At leptotene, a subset of loops becomes tethered to the underlying chromosomal axis and Spo11-induced meiotic DSBs then occur within these complexes (Padmore *et al*, 1991; Blat *et al*, 2002; Panizza *et al*, 2011; Acquaviva *et al*, 2013; Sommermeyer *et al*, 2013). After a DSB is made, one DSB single-end searches for a homologous partner and engages into recombination (Kim *et al*, 2010; Lam & Keeney, 2015). A subset of these nascent pairing intermediates (presumably D-loops) are designated to mature into crossovers (COs), while the others are quickly resolved as non-crossovers (NCOs) (Allers & Lichten, 2001; Hunter & Kleckner, 2001). The complex network of proteins that nucleates at CO-designated DSBs sites eventually forms the SC (Zickler & Kleckner, 1999; Henderson & Keeney, 2004).

Evidences support transient and/or sparse enrichment in homolog contacts during mitosis (Burgess *et al*, 1999), but pairing during meiosis occurs robustly on a much larger scale, along entire chromosomes. Homologs have therefore to meet each other in space, which implies a dynamic reorganization of the overall genome and disentanglement between DNA molecules (Zickler & Kleckner, 2016). This process can be accompanied and/or facilitated by dynamic movements of chromosomes and lead to telomere clustering at the zygotene stage (bouquet stage; Zickler & Kleckner, 2016) or other forms of movements mediated by chromosome ends directed by cytoskeletal components through direct association across the nuclear envelope (Koszul & Kleckner, 2009). These events have mostly been described using imaging techniques or, when it comes to the analysis of the underlying molecular events, through site-specific assays. Notably, Dekker, Kleckner, and coworkers pioneered the analysis of meiosis using chromosome conformation capture (3C) in their seminal study (Dekker *et al*, 2002). Using restriction site polymorphisms to distinguish the maternal and paternal versions of a locus along chromosome III, they showed that 3C was able to capture homolog pairing, as well as centromere declustering. However, the higher-order organization surrounding the recombining loci and over the entire genome remains unexplored.

In order to investigate the behavior of two homologous chromosomes, we designed and assembled a dedicated synthetic genomic region (Koszul, 2016; Richardson *et al*, 2017), with an increased resolution for 3C-based experiments. As a proof of concept of this strategy, we describe here a redesigned 144-kb region (called Syn-HiC) of *Saccharomyces cerevisiae* yeast chromosome IV and track its behavior during the first stages of meiotic prophase.

## Results

### Design and assembly of the Syn-HiC region

Designer chromosome Syn-HiC closely resembles the native chromosome with respect to genetic elements (see Materials and Methods for details and Dataset EV1 for the sequence), but was "designed" to yield high resolution and high visibility in 3C experiments by providing nearly equally spaced restriction sites (Fig EV1). The RSs of four different enzymes were removed with point mutations from the native sequence of the *S. cerevisiae* SK1 background and subsequently reintroduced within the sequence at regularly spaced positions (400, 1,500, 2,000 and 6,000 bp for DpnII, XbaI, HindIII, and NdeI, respectively; Figs 1B and EV2; Table 1). As shown on Fig 1C, the DpnII and HindIII RF size distributions in the redesigned Syn-HiC region display sharp contrasts to the native genome-wide distributions that are skewed toward smaller fragments. When possible, coding sequences were targeted preferentially and modified using synonymous mutations. We identified a 150-kb window on chromosome IV for which the uniformity of RF lengths was maximized while the number of potentially deleterious base changes was minimized (the final choice for the region also takes into account sequence annotation and was guided by specific interests of the end-user). From this design, DNA building blocks were purchased as 3-kb fragments and assembled in yeast BY (S288C) and SK1 background strains as described (Muller *et al*, 2012; Annaluru *et al*, 2014) (Materials and Methods; Table 2; Fig EV3A and B). Sequencing confirmed that 140 kb within the targeted region, encompassing open reading frames (ORFs) YDR127w to YDR196c, was replaced by the redesigned sequence and that all the mutations were introduced at the correct positions corresponding to a total of ~2% divergence with the reference genomes (3,229 bp out of 145,000 bp). Analysis of the growth profile did not reveal significant effects of the modifications introduced in the Syn-HiC region compared to the isogenic parental strain (Fig EV3C). A transcriptional profiling of the Syn-HiC region was performed to identify potential changes in gene expression between the Syn-HiC and parental strains. RNA-seq experiments were performed in triplicate for each strain and compared using DESeq2 (Love *et al*, 2014). Six non-dubious ORFs (out of 69) were found to be differentially up- or down-regulated within the Syn-HiC region (Fig EV3D), suggesting that some point mutations introduced within non-coding regions can affect transcription. Nevertheless, these relatively minor changes were unlikely to affect the conserved, robust 3D patterns of budding yeast chromosomes (Duan *et al*, 2010;

**Table 1. Mutations necessary to remove and generate new sites along chromosome IV 700,000::850,000 window in the SK1 background.**

|  | Deletion | New sites |
|---|---|---|
| HindIII | 58 | 61 |
| NdeI | 34 | 23 |
| XbaI | 25 | 76 |
| DpnII | 442 | 310 |
| Total | 559 | 470 |

**Table 2.  Genotype of *Saccharomyces cerevisiae* strains used in this study.**

| Name | Genotype | Background | Source |
|------|----------|------------|--------|
| YRSG181 | *MATalpha ura3Δ0, leu2Δ0, his3Δ1, lys2Δ0, IV(715448-845757)::synIV(715448-845757 LEU2)* | S288C | This study (YKL050 in Lazar-Stefanita et al, 2017) |
| YRSG189 | *MATa, ura3, lys2, ho::LYS2, leu2-R, arg4-nsp,bgl, IV(715448-845757)::synIV(715448-845757 URA3)* | SK1 | This study |
| YRSG190 | *MATa/MATalpha ura3/ura3, lys2/lys2, ho::LYS2/ho::LYS2 leu2-K/leu2-R, arg4-nsp,bgl/arg4-nsp,bgl, IV(715448-845757)::synIV(715448-845757 URA3)/IV(715448-845757)* | SK1 | This study |
| YRSG154 | *MATa/MATalpha ura3/ura3, lys2/lys2, ho::LYS2/ho::LYS2 leu2-K/leu2-R, arg4-nsp,bgl/arg4-nsp,bgl, IV(715448-845757)::synIV(715448-845757 URA3)/IV(715448-845757), ndt80::kanMX/ndt80::KanMX* | SK1 | This study |
| ORT4601 | *MATalpha, ura3, lys2, ho::LYS2, leu2-K, arg4-nsp,bgl* | SK1 | Sollier et al (2004) |

Mercy *et al*, 2017), and we moved forward with the analysis of the folding of the redesigned region.

## Cis-contact pattern of the Syn-HiC region

To estimate the quality of Hi-C data in the Syn-HiC region, Hi-C experiments were performed in parallel on G1 synchronized cells of strain carrying the Syn-HiC redesigned chromosome (YRSG181) and the corresponding parental strain (BY4742) using DpnII and HindIII (Materials and Methods). The raw DpnII contact map of chromosome IV exhibits a remarkably "smooth" pattern within the redesigned region compared to the native flanking regions (Fig 1D). The read coverage over the region also exhibits a dramatic and compelling change, with a more homogeneous and regular distribution in the synthetic regions for both enzymes compared to a heterogeneous distribution in the native sequence, where clusters of restriction sites result in an increased capture frequency of neighboring fragments (Fig 2A and B). To quantify the improvement in the Syn-HiC region, we compared the contact signal in the Syn-HiC region with the signal over the same region obtained in the parental strain using the same number of aligned read pairs and identical bins of various sizes (Fig 2C and D; see also Fig EV4). At the smallest resolution tested (600 bp for DpnII and 2,400 bp for HindIII), the parental contact map exhibited numerous blind regions with no detectable contacts (empty bins), in sharp contrast to its synthetic counterpart (Fig 2C and D). When fragments were aggregated in bins of increasing sizes (hence, resulting in a loss of resolution), these blind regions gradually disappear, although the heterogeneity of the data remains consistently higher in the parental (wt) strain compared to Syn-HiC strain, as shown by the increased span of the color scales of the parental maps.

In order to further quantify this heterogeneity, we computed the cumulative distributions of the number of contacts between bins separated by a given genomic distance *s* (bp) in the Syn-HiC region and in its parental counterpart for DpnII and HindIII (Fig 2C and D, respectively). The redesigned region systematically exhibited more homogeneous contact counts and narrower distributions than the native region, both at short ($s = 2 \times$ bins sizes; Fig 2C and D middle panels) and at longer distances (Materials and Methods; Fig EV5). Some of the bins in the native region remain almost invisible to the assay as a result of the heterogeneity in RF distribution (blue squares in Fig 2C and D, middle panels). We computed the coefficient of variation (CV) (i.e., standard deviation/mean) of these distributions for multiple values of *s*. We use this value as an indication of the signal-to-noise ratio (Fig 2C and D, right panels). Interestingly, we found that even for large bins, the CV is significantly and consistently smaller in the synthetic region, again indicating improved resolution. These results also clearly illustrate the advantage of using a frequent cutter (DpnII versus HindIII) restriction enzyme with respect to resolution since the distribution of contact counts between bins remains much more spread with HindIII than with DpnII, especially for native sequences (Fig 2B).

## Statistical analysis of Hi-C contact data

Having regularly spaced restriction sites along the Syn-HiC region opened up the possibility to tackle pending questions revolving around the definition of the resolution in a Hi-C experiment. The sequencing step of a Hi-C library corresponds to the random sampling of all ligation events generated during the experiment. If we suppose that each pair of loci at a given genomic distance $s \pm \Delta s$ has a strictly equal probability *p* to be drawn, the outcome of this draw is expected to follow a Poisson distribution. For real data, we expect that other factors influence this probability. These can result from experimental limitations (such as restriction fragment size) or from the measured object itself (i.e., the average 3D folding of chromatin). When the number of events becomes large enough, the differences between the re-ligation probabilities of different loci will start to kick in and the distribution of contacts should switch from a Poisson to a Gaussian behavior, assuming that the contact probabilities associated with these factors follow a Gaussian distribution. The standard deviations ($\sigma$) of those two distributions (Poisson and Gaussian) scale respectively as the square root of the mean ($\sqrt{\mu}$) or as the mean ($\mu$). We took advantage of the fact that the overall contact number $\mu$ decreases with increasing genomic distances (s) to check whether these relations between $\mu$ and $\sigma$ hold for real data (Fig 3A).

We first aggregated the results from 12 Hi-C experiments performed in G1/early S phase and with or without the Syn-HiC region (data from Lazar-Stefanita *et al*, 2017). In the following, we focused our analysis on the Syn-HiC region. For five different distances (Fig 3A), the contact distributions computed from each restriction fragment pair were re-scaled by their mean and superposed on the same plot (Fig 3A, inset). The collapse of these re-scaled distributions clearly indicates a Gaussian behavior (i.e., the mean scales as the standard deviation). We next explored the relation between $\mu$ and *s* over a wider range of values. We used data for increasing values of *s*, corresponding to lower $\mu$, as well as data

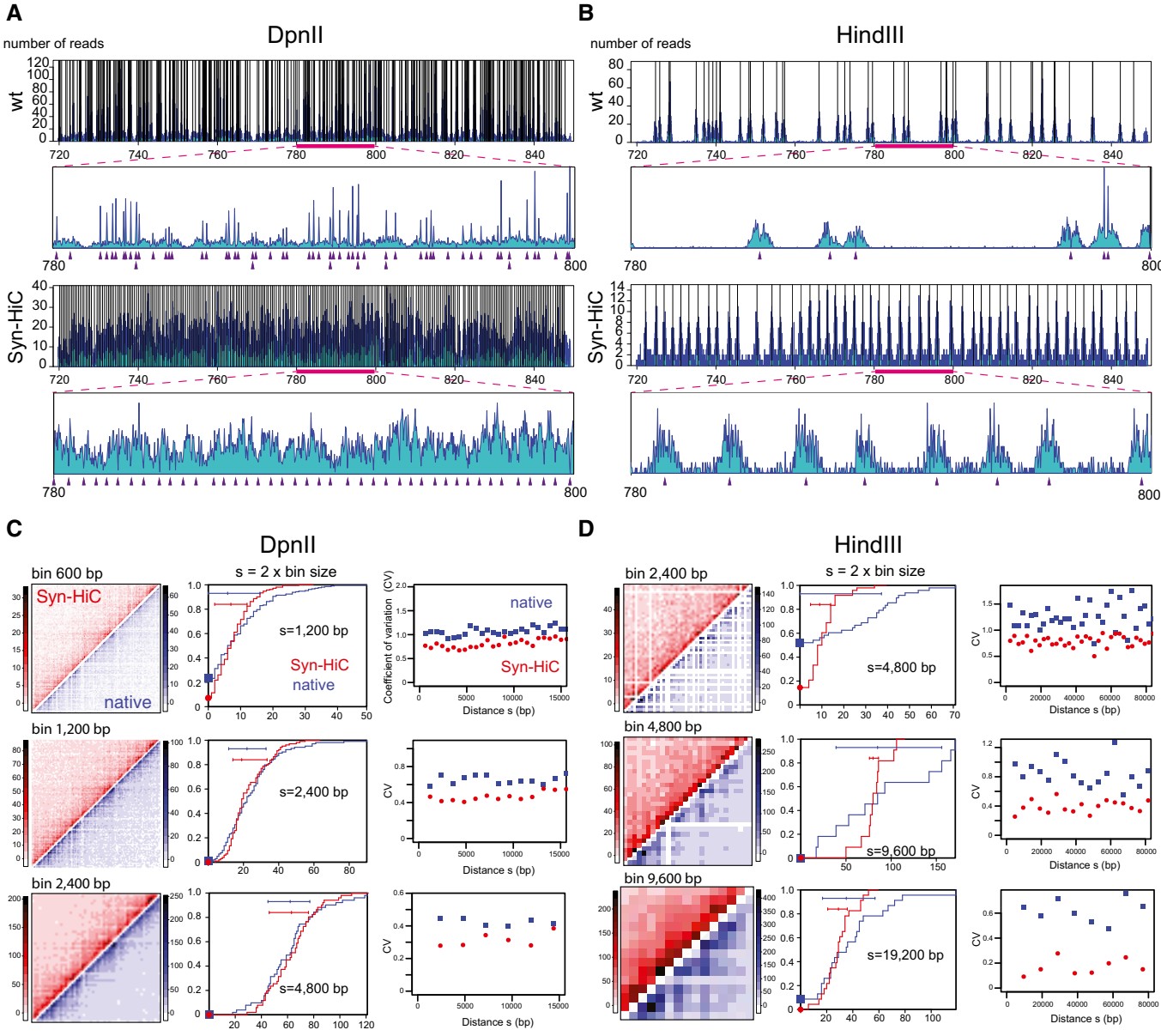

**Figure 2. Distribution of contacts along the Syn-HiC and native regions.**

A, B  Sequence read coverage from independent Hi-C experiments performed with DpnII (A) and HindIII (B) restriction enzymes in haploid Syn-HiC and native strains. The same number of reads was aligned against the synthetic chromosome IV region and its native counterpart. For each experiment, an overview of the entire region is shown in the top panel, with the magnification of a 20-kb window presented below (corresponding to the region underlined with a pink bar). The black vertical lines and the purple triangles point at restriction sites positions in the top global overview and the bottom magnification panels, respectively. Note that the scale of the y-axis illustrates the heterogeneity of the coverage, particularly strong in the native region.

C, D  Analysis of the contact counts along the Syn-HiC region for DpnII (C) and HindIII (D). Left panels: Syn-HiC (in red) and chromosome IV native counterpart (blue) Hi-C contact maps. For each experiment, three different fixed bin sizes were analyzed (600, 1,200 and 2,400 bp for DpnII; 2,400, 4,800 and 9,600 bp for HindIII). Middle panels: cumulative distribution of the number of contacts between pair of bins separated by a given distance (in bp) $s = 2 \times$ bin size (x-axis: read number, y-axis: cumulative probability). Horizontal bars: 25th to 75th percentile. Vertical dot on the horizontal bar indicates the median. Right panels: distribution of the coefficient of variation (CV) as a function of $s$.

aggregated over multiple experiments and data from capture experiments, corresponding to higher values of μ (Fig 3B). When plotting the values of $s$ for different μ, we found that both in the native and in the Syn-HiC sequence context, there is a crossover from the Poisson to the Gaussian distribution (indicated by the red and blue lines, respectively) and that the standard deviation for the Syn-HiC

experiment is lower than in the wt counterpart for all the values of μ, as expected from the analysis done in Fig 2. Interestingly, the two transition points can tell us about the importance of the bias of having uneven restriction fragments compared to other biases and/or biologically relevant variations. In the native case, as soon as each pair of fragments receives on average one read, the

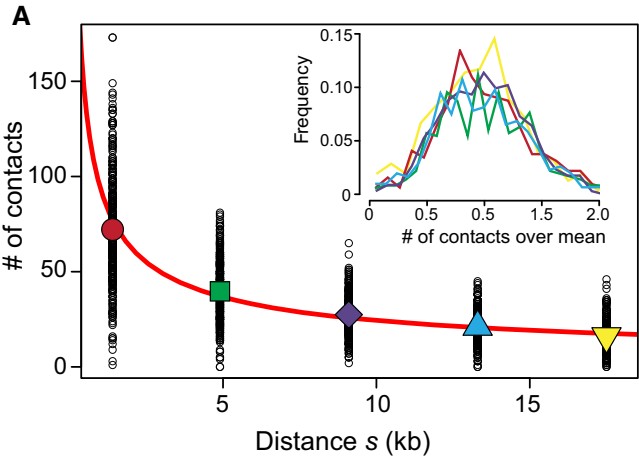

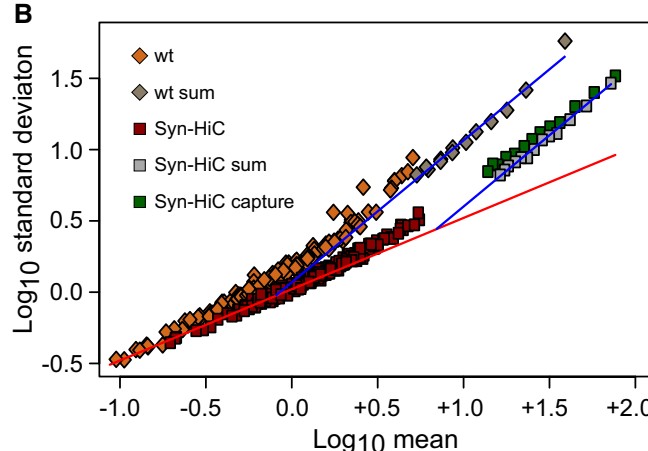

**Figure 3.   Relation between standard deviation and mean for each pair of fragments located at similar genomic distances $s \pm \Delta s$.**

A   The mean number of (raw) contact of restriction fragments (big symbols) at a fixed range of distances ($s \pm 350$ bp, with $s = 1,400 \pm 350$ bp in red; $4,900 \pm 350$ bp in green; $9,100 \pm 350$ bp in purple; $1,330 \pm 350$ bp in cyan; and $17,500 \pm 350$ bp in yellow) decreases as a power law (red line) with exponent $-0.6$. Small circles represent the contact number of individual pairs of restriction fragments. Inset: distributions of contact numbers re-scaled by their means.

B   The relation between the variance and the mean of these distributions undergoes a transition from Poisson to Gaussian. wt and Syn-HiC data were obtained by aggregating datasets respectively from Mercy *et al* (2017) and Lazar-Stefanita *et al* (2017). Capture-C data are from this study. The red line corresponds to the theoretical Poisson behavior, whereas the blue line corresponds to the theoretical Gaussian behavior fitting the CV from data.

distribution of counts switches to Gaussian, indicating that the strong bias induced by the uneven length of RF already kicked in. In the case of the Syn-HiC construct, where this bias is absent, one needs to aggregate 10 reads per fragment pairs to start seeing variations among fragment pair re-ligation frequencies and switch to the Gaussian behavior. This highlights the strong effect of fragment length biases and justifies the use of large bins which will encompass many fragments as well as normalization procedures in all Hi-C experiments. For any genomic distance $s$, the value of $\mu$ can arbitrarily be increased by increasing the bin size, at the expense of the resolution.

The existence of the transition between Poisson and Gaussian behaviors allows to propose a rigorous way to determine the resolution of a Hi-C experiment by choosing a bin size which corresponds exactly to the transition point. According to this the resolution of a Hi-C experiment can only be defined for a given genomic distance $s$.

**Analysis of genome organization during meiosis prophase**

The Syn-HiC chromosome was designed and assembled with the aim to investigate the folding and interplay of homologous chromosomes during the meiotic cell cycle (Fig 4A). A diploid SK1

**Figure 4.   Individualization of chromosomes during prophase.**

A   Schematic representation of the structural changes affecting homologs during meiotic prophase. Sister chromatids organize as arrays of loops along each homolog axis (green lines) after replication. At leptotene, DNA double-strand breaks (DSBs) occur. During zygotene, homologs come together partially and the synaptonemal complex (SC, gray) originates at DSBs and centromeres. At pachytene, homologs are fully synapsed and undergo vigorous motion (purple arrows).

B   Meiotic progression, as measured by completion of the first meiotic division (MI, orange line) and the second meiotic division (MII, red line). %nuc: percent of total nuclei analyzed. SPM: sporulation medium. Red circles: time points sampled from this time course.

C   Normalized contact maps of synchronized populations of cells after 0 (bottom left) and 4 (top right) hours in SPM. The 16 yeast chromosomes are displayed atop the maps. Purple arrowheads: telomeric contacts. Green arrowheads: inter-centromeric contacts. The inset on the bottom left displays a magnification of chromosome XV from both maps in a *vis-à-vis* disposition. Dotted lines delimit the centromeric region subject to a "polymer brush" effect.

D   3D representations of contact maps at 3 and 4 h and in pachytene.

E   Log-ratio of different pairs of contact maps ordered as a function of meiotic progression (i: 3 h versus 0 h; iii: 4 h versus 3 h; v: 6 h versus 4 h; vii: ndt80-arrest versus 6 h). The blue-to-red color scale reflects the enrichment in contacts in one population with respect to the other (log2). The insets ii, iv, and vi display magnifications of chromosomes XV and XVI from the above maps. Under each magnification, the dotted lines point at the increase in contacts made by either the centromere (stars) or a telomere (T) of each chromosome in the newest condition compared to the oldest one.

F   Illustration of the large structural changes affecting chromosomes during prophase and supported by the ratio maps displayed in (E). For clarity, no chromatin loops are represented and only the purple chromosome is represented with two homologs. The dotted arrows represent the telomere-driven meiotic movements that initiate at zygotene.

G   Boxplots representing the variation in number of normalized contacts for centromeres (left) and telomeres (right) between 0, 3, and 4 h in SPM. For each dataset the box represents the 25th and 75th percentiles. Horizontal red line: median. The whiskers default is to cover 99.3 percent of the data (x: outliers). Variations between $t = 0$ h replicates, 3 h and 0 h, and 4 h and 0 h were computed. A relative Wilcoxon test supports a decrease in centromere contacts at $t = 3$ and 4 h compared to $t = 0$ h.

H, I   Average intra-chromosomal normalized contact frequency p between two loci with respect to their genomic distance s along the chromosome (log-log scale; $p(s)$) during (H) mitotic G1 (brown curve; three replicates; error bars correspond to the standard deviation; Lazar-Stefanita *et al*, 2017), meiotic $t = 0$ h (two replicates), 3, 4 and 6 h, and (I) pachytene (ndt80$\Delta$-arrested) and mitotic metaphase (cdc20-arrested) cells.

strain (YRSG190), carrying the Syn-HiC region on one homolog and its native counterpart on the other (but isogenic for the rest of the genome), was either processed into a synchronized meiotic culture or arrested in pachytene using a deletion mutant of *NDT80*, a transcription factor encoding gene required for late pachytene progression and CO formation (Xu *et al*, 1995; Materials and Methods). The synchrony of meiotic progression was assessed by monitoring meiotic replication by FACS analysis and

the two meiotic divisions by SYBR Green staining. Cells that completed anaphase I and anaphase II contain two or four stained bodies, respectively. By 8 h, 70% of the cells have passed anaphase II, showing that most cells have completed meiosis (Fig 4B; Hunter & Kleckner, 2001). Hi-C contact maps were generated for cells sampled at 0, 3, and 4 h in sporulation medium (SPM), corresponding to wt mitotic, mostly early-zygotene, and mostly early-pachytene cells, respectively (Figs 4B and C, and

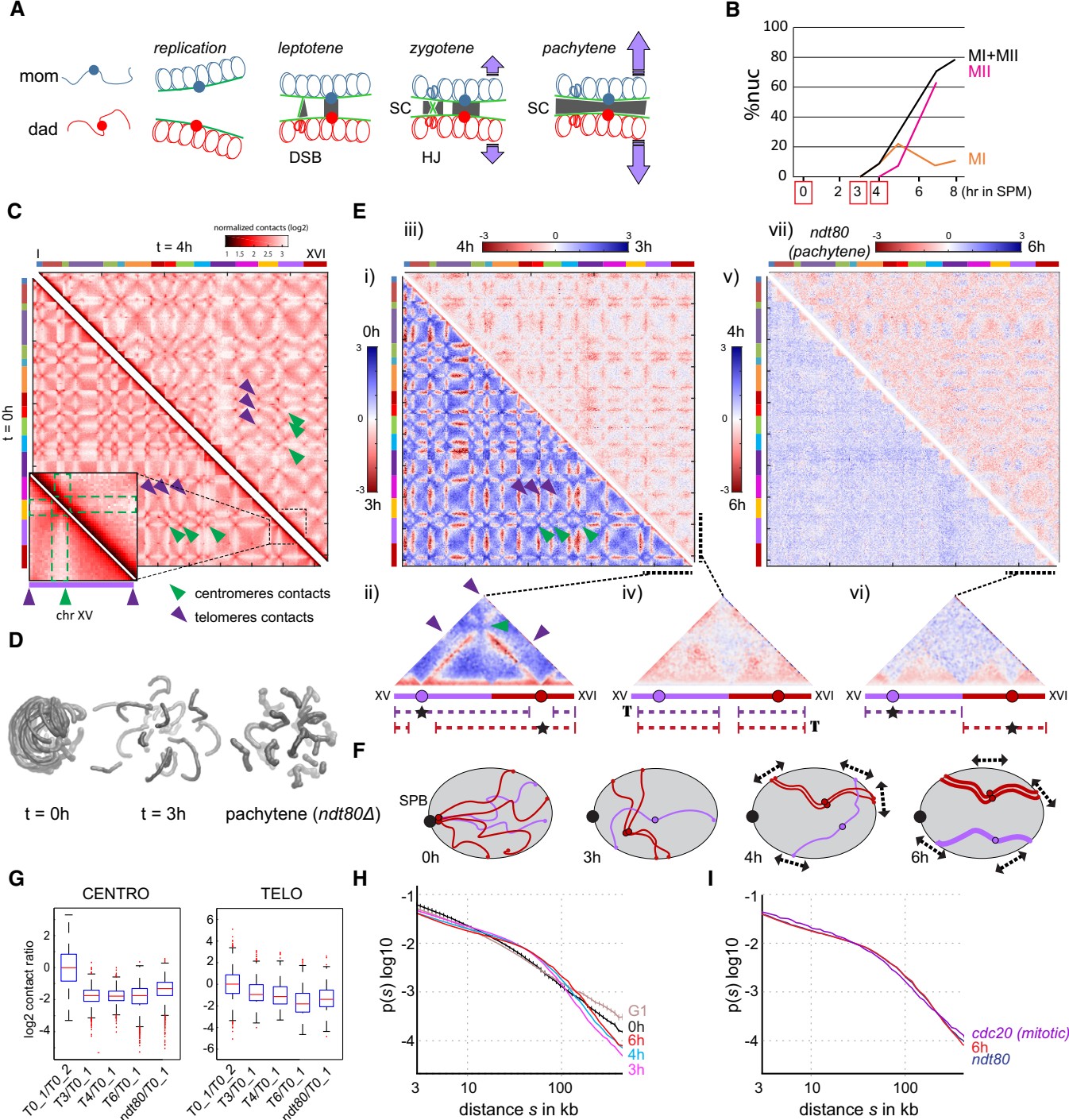

**Figure 4.**

EV6A; bin: 2.5 kb). The fact that a fraction of the population passed anaphase I at 4 h shows that the synchronization is not perfect. Contact maps of pachytene in *ndt80Δ*-arrested and wild-type cells at 6 h were also generated (Fig EV6A). Individual 2D maps can also be represented as 3D structures to illustrate the transformation of chromosomes into well-individualized entities throughout prophase (Fig 4D; Lesne *et al*, 2014). Differences between the Hi-C datasets were assessed by computing the element-wise log-ratio between the normalized contact maps (bin: 25 kb; Figs 4E and EV6B; Materials and Methods). Under this representation, the color scale of the map reflects the local variations in contact frequencies between the two contact maps. Overall, our results illustrate and recapitulate known features of meiotic prophase.

### Declustering of centromeres

First, a loss of inter-centromeric contacts was readily apparent at 3 h compared to the pre-replication stage (green arrowheads; Fig 4C, D and G; see also Fig EV6A). This result reflects the rapid declustering of centromeres that accompanies the entry into meiotic prophase also observed in microscopy and 3C (Trelles-Sticken *et al*, 1999; Dekker *et al*, 2002). In addition to the loss of discrete centromeric contacts, the contact ratio map reveals the abolition of the "polymer brush" effect (Fig 4E i and magnification of chromosomes XV and XVI on Figs 4E ii and EV6B) which insulates centromeres and their flanking chromosomal regions from the rest of the chromosome arms (compare the *cis*-contacts made by chromosome XV centromere at 0 and 4 h: the green dotted area on Fig 4C; Duan *et al*, 2010; Wong *et al*, 2012). As a result, these pericentromeric regions now behave similarly to the rest of the chromosome. Full-length chromosomes therefore display a much more homogeneous pattern (without the typical cross-shaped pattern seen in *t* = 0 h Hi-C maps; Fig EV6A). Not only declustered pericentromeric regions do not "see" each other as much, but they also become more prone to contact all other portions of the genome, which translates as (red) dashed lines of enriched contacts on the ratio map (Fig 4E i, ii). The release of centromere constraint and their ability to contact the rest of the genome compared to the pre-meiotic/mitotic condition are schematically represented in Fig 4F.

### Telomere dynamics

No significant enrichment in contacts between telomeres was observed at the different stages (purple arrowheads in Fig 4C, E and G). The transient apparition of a telomere, bouquet-like cluster in a subset of cells described at the zygotene stage through imaging (Trelles-Sticken *et al*, 1999) was therefore not captured by the Hi-C approach. On the opposite, telomeres exhibit a gradual loss of contacts over time. These changes were accompanied by other modifications in their contact patterns as observed on the ratio of Hi-C maps between time 3 and 4 h (Fig 4E iii, iv). At 4 h, whereas they lose contacts with each other, telomeres appear to contact more frequently chromosome all along their lengths. This signal is in agreement with the initiation of vigorous telomere-mediated chromosome movements at this stage that continue until the first division, and supports a model in which movements promote the resolution of interlocked chromosomes by their ends (Koszul *et al*, 2008; see Discussion; Fig 4F).

### Meiotic chromosomes' contact patterns resemble mitotic metaphase chromosomes'

The contact ratio maps display a strong and continuous increase in short-range, intra-chromosomal contacts during prophase compared to mitotic cells that culminates at *t* = 6 h (Figs 4E v, vi and EV6A and B; see also Fig 5A), when cells display contact patterns very similar to *ndt80Δ*-arrested pachytene cells (Fig 4E vii). This change can be assessed quantitatively by computing the contact probability *p* as a function of genomic distance *s* of all chromosome arms (Lieberman-Aiden *et al*, 2009; Naumova *et al*, 2013; Lazar-Stefanita *et al*, 2017; Schalbetter *et al*, 2017). The *p*(*s*) curves were computed for two pre-meiotic replicates (0 h), three mitotic G1 and from pre-meiotic cells released into SPM for 3, 4, and 6 h (Fig 4H). *P*(*s*) of pachytene (*ndt80Δ*-arrested) cells was also computed, displaying a pattern very similar to wild-type cells after 6 h in SPM, as well as to mitotic metaphase (*cdc20*-arrested) cells (Schalbetter *et al*, 2017) (Fig 4I). The *p*(*s*) curves of meiotic cells (3, 4, and 6 h and *ndt80Δ*-arrested) display sharp differences compared to pre-meiotic and G1 cells. First, contact frequencies increase between loci positioned 20 to 50 kb apart, with a peak around ~50 kb. For longer distances, the contact frequencies sharply decrease, although a small increase in long-range contact can be observed as cells progress toward pachytene (red curve versus pink curve). This result is compatible with the structuring of the chromosome into arrays of loops which would favor such medium-range contacts while disfavoring long-distance contacts. Mitotic metaphase and meiotic prophase chromosomes are strikingly similar, pointing at a similar internal structure (Fig 4I). Altogether, these observations suggest that chromosomes fold into a structure that favors contacts under a certain distance during meiotic prophase, compatible with the formation of mitotic-like loops, while disfavoring longer range contacts.

## Visualization of meiotic loops using Hi-C

We took advantage of this pilot study to investigate the presence of loops and address their demarcation by cohesin. First, we investigated whether Rec8-mediated loops were visible on the different meiotic chromosomal contact maps by plotting the Rec8 binding regions along chromosome lengths (chromosomes V and VI in Fig 5A; see also Fig EV6A and C) (Ito *et al*, 2014). The pre-meiotic map (*t* = 0 h) displays a relatively regular contact pattern similar to those observed during G1 (Lazar-Stefanita *et al*, 2017). On the other hand, triangular darker shapes corresponding to regions of enriched contacts appear along the chromosomes of the meiotic maps, reminiscent of the topologically associated domains (TADs) observed in metazoans, though smaller (Dekker & Mirny, 2016; Yu & Ren, 2017). The boundaries between these domains correlate with the position of Rec8-enriched regions (Fig 5A; Glynn *et al*, 2004; Ito *et al*, 2014; Sun *et al*, 2015). In addition, discrete spots corresponding to enriched contacts between pairs of distant regions appear at variable distances from the main diagonal (Fig 5A, yellow dotted lines on the *ndt80Δ*-arrested pachytene map). This signal provides direct molecular evidence for the presence of chromatin loops of various sizes along the chromosomes during meiosis prophase.

The bases of these loops appear to correlate with pairs of adjacent Rec8 deposition sites (blue lines, Fig 5A). In addition, a weak "looping" signal bridging non-adjacent Rec8 positions was also distinguishable on the maps (black dotted lines on the

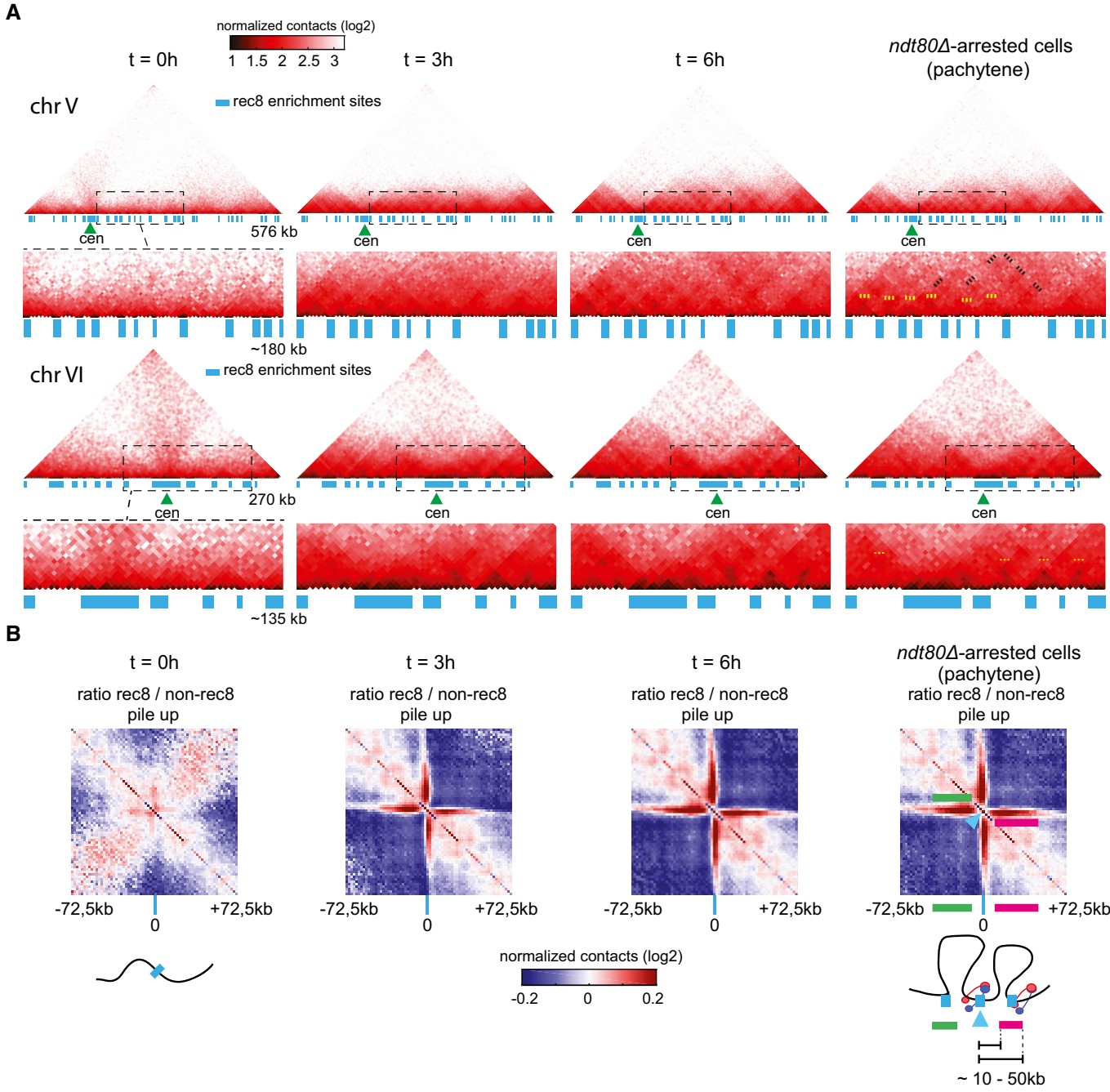

**Figure 5. Characterization of meiotic chromatin loops.**

A  Normalized contact maps (bin size: 5 kb) of chromosomes V and VI after 0, 3, and 6 h in SPM and of pachytene (*ndt80Δ*-arrested) cells. Magnification of ~180-kb and ~135-kb regions (dotted boxes) is displayed under chromosomes V and VI, respectively. The blue rectangles point at bins enriched in Rec8 protein. Green triangles: centromere position. The dotted yellow and black lines point at contact enrichment between adjacent and non-adjacent Rec8 deposition sites, respectively.

B  Ratio between the aggregated normalized contact maps made by 145-kb intra-chromosomal windows (2.5-kb bins) centered on Rec8-enriched bins (blue line on the bottom axis) and non Rec8 bound bins. Blue color shows a depletion of contacts in the random maps, whereas the red signal points at an enrichment in contacts in the maps centered on Rec8-enriched bins. In pachytene map, green and pink lines point at the looping signal between the upstream and downstream flanking regions of the Rec8-enriched bin (magenta arrowhead). A schematic representation of the disposition of the chromatin inferred from the pattern of the ratio map is displayed for pre-meiotic (0 h) and pachytene (*ndt80Δ*-arrested) cells.

*ndt80Δ*-arrested map, Fig 5A). To further characterize the loops, we compared the aggregated intra-chromosomal contacts made by Rec8-binding sites to random positions in pre-meiotic (0 h) and

meiotic (3, 4, and 6 h and *ndt80Δ*) cells. The ratio between the average contact maps (2.5-kb bins) centered on Rec8 binding sites with the average maps computed on sites which are not bound by Rec8

displays no significant enrichment in pre-meiotic (*t* = 0 h) conditions (Fig 5B). However, the same analysis gives a strikingly different pattern during prophase, with the Rec8-bound sites (magenta marks) clearly delimiting distinct domains (Fig 5B; see also Fig EV6C). These sites now display enriched contacts with DNA regions positioned on average between ~10 and 50 kb both upstream and downstream along the chromosome (underlined by green and purple lines on the *ndt80Δ*-arrested cells panel, respectively), but much less contacts with regions positioned closer (between 2.5 and 10 kb). The elongated shape of the enriched contact signal supports a heterogeneity in the size of the loops over the genome, in agreement with the heterogeneity of distances separating Rec8 deposition sites (Glynn *et al*, 2004).

Overall, this experiment represents a direct evidence of the establishment of chromatin loops of various sizes during yeast meiotic prophase. The bases of these loops involve for the most part

adjacent Rec8 deposition sites, but contacts involving non-adjacent sites are also visible on the pachytene maps.

## Homolog–homolog contacts between the Syn-HiC and native regions

To characterize the inter-homolog contacts between the synthetic and the native chromosome IV regions, an enrichment step for this region was performed using a Capture-C strategy at 0, 3, and 4 h (Hughes *et al*, 2014). This led to a ~20-fold increase in read counts from the chromosome IV region of interest that were used to generate contact maps (Fig 6A, left panels). The computation of the *p(s)* within each of the two homologous regions, which are distinguishable from each other thanks to the SNPs introduced in the synthetic design, reveals a similar trend than the ones computed over the whole genome for the three time points (Fig 6B), showing that the Syn-HiC design does not

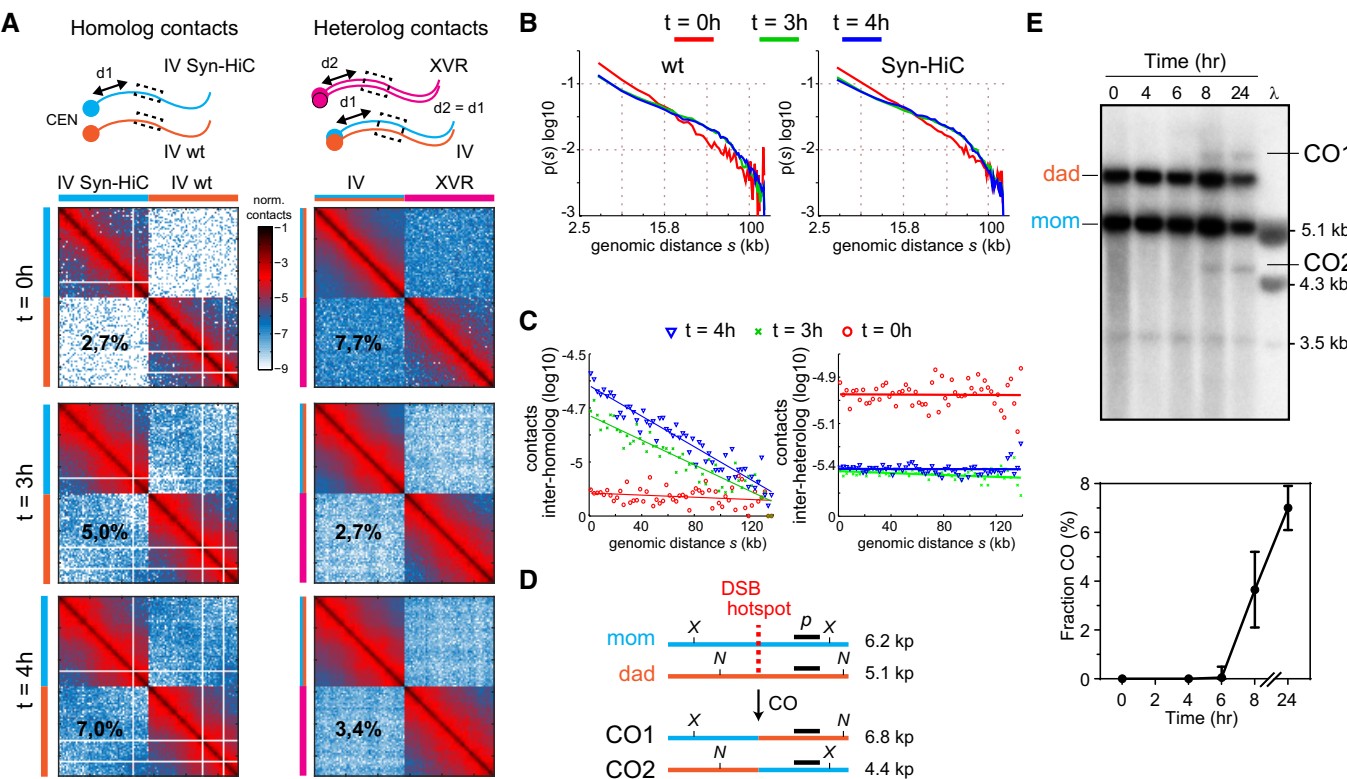

**Figure 6.  Homolog pairing.**

A   Normalized contacts between the Syn-HiC region and its native homolog (left panels) and between the aggregated chromosome IV region and one of similar size positioned at an equal distance (d1 = d2) from the centromere but on another chromosome (right panels). The percentages on each column reflect different measurements. For the inter-homolog panels, they represent the number of read pairs bridging homologs (*trans* contacts) divided by the total number of pairs aligning within the regions. For inter-heterologs, the percentage represents the amount of read pairs bridging the two distinct regions, divided by the total number of pairs aligning within the distinct regions. This illustrates the gradual chromosomal individualization.

B   Normalized frequency of contacts as a function of genomic distance within the Syn-HiC and its native counterpart for *t* = 0, 3, and 4 h.

C   Left panel: inter-homolog contacts. The mean contacts between pairs of loci with one locus within the Syn-HiC region and the other on the homologous wt window were computed at *t* = 0, 3, and 4 h, and for increasing genomic distances separating these positions (x-axis). Right panel: similar plot for the inter-heterolog windows of panel (C).

D   Crossing-over between the Syn-HiC region and the native counterpart: schematic representation of how the redesigned restriction pattern allows characterization of CO events using a probe (black line *p*) at a DSB hotspot (Materials and Methods).

E   Top panel: meiotic recombination of cells progressing into meiosis and analyzed using a restriction assay similar to Hunter and Kleckner (2001). Parental homologs, "dad" and "mom", and COs are distinguished on Southern blots via restriction site polymorphism. Bottom panel: CO as percent of total hybridizing signal with time after transfer to sporulation medium (error bars: standard deviation from 2 independent experiments).

affect chromatin folding. Inter-homolog contacts increase over time, from 2.7% of reads before replication to 7.0% at 4 h into meiosis. In sharp contrast, contacts between these two homologous regions and a heterologous region positioned at a similar distance from the centromere (on chromosome XVR) decrease over time (Fig 6A, right panels). These results underline the insulation of heterologous chromosomes as cells enter meiotic prophase, while homologous chromosomes synapse (see also Fig 4C and E).

The two homologs become loosely juxtaposed, as shown by the weak diagonal signal that gradually appears over time in the inter-homologous contact maps (plotted in Fig 6C). To verify whether the Syn-HiC design and its ~2% sequence divergence with the native region affect meiotic recombination, we monitor CO formation taking advantage of the restriction site polymorphisms (Fig 6D). COs were detected at the *CCT6* meiotic DSB hotspot present within the region, showing that meiotic recombination can proceed in this genetic environment (Fig 6E). More analyses remain to be performed to verify that this is the case over the entire region, but this experiment shows that the present approach has the potential to track recombination events concomitantly to the higher-order architecture of the chromosomes.

# Discussion

Like all genomic approaches, Hi-C is unable to track the 3D organization of large, repeated sequences, as well as homologous chromosomes in isogenic strains. Here, we show that redesigning and assembling a large (144 kb) chromosomal segment (Syn-HiC design) in yeast that incorporates regularly spaced restriction sites alleviates this limitation. The polymorphisms introduced into the redesigned region allowed to distinguish it from its native counterpart, in an otherwise isogenic background. Not only do these polymorphisms allow tracking both homologs, but the regular spacing of restriction sites also improves the quality of Hi-C data, which in turn provides a new definition of the resolution in Hi-C experiments.

### An array of heterogeneous chromatin loops during meiosis

Here, we took advantage of the Syn-HiC design to track the large and dynamic structural changes of chromosomes during the meiotic prophase using Hi-C. Replicated meiotic chromosomes reorganize as arrays of chromatin loops anchored along a chromosomal axis. In budding yeast, the limited resolution of cytological approaches failed to yield detailed characterization of the structuring mechanism and of the loop features (size, heterogeneity, contacts) (Møens & Pearlman, 1988). ChIP analysis of chromosome axis-structuring components (Hop1, Red1, and Rec8) revealed periodic enrichment sites whose correct localization depends on the meiosis-specific cohesin (Klein *et al*, 1999; Panizza *et al*, 2011). Put in relation to cytological data, this periodic recruitment suggested a typical loop length of ~20 kb (Møens & Pearlman, 1988; Zickler & Kleckner, 1999, 2016; Blat *et al*, 2002). The Hi-C data reveal contact patterns compatible with the formation of loops of different sizes ranging from ~10 to 50 kb (Fig 5B), whose bases overlap mostly with adjacent, but also occasionally with non-adjacent, binding sites of the meiosis-specific cohesion subunit Rec8. Chromatin-loop extrusion mediated by cohesin/condensin has been proposed to actively condense chromosomes

during mitotic prophase (Goloborodko *et al*, 2016; Schalbetter *et al*, 2017; Gibcus *et al*, 2018). In addition, the distribution of the distances separating mitotic cohesin-enriched sites ranges between 5 and ~50 kb, a distribution very similar to the meiotic loop sizes (Glynn *et al*, 2004). As a result, a similar loop extrusion mechanism could account for the condensation of meiotic chromosome during early prophase (Fig 7). The contacts between non-adjacent Rec8-binding sites suggest (i) that the same sites are not systematically used in all the cells and/or (ii) that occasionally a larger loop is formed by merging two smaller ones and/or (iii) compaction may bring closer distant loci. This heterogeneity should be further characterized using Hi-C derivatives such as single cell or C-walk that discriminate subpopulations of chromosome structure in a population of cells (Nagano *et al*, 2013; Olivares-Chauvet *et al*, 2016). Why loops would be more visible in meiosis compared to mitosis at the same Hi-C resolution remains to be characterized, but it is tempting to suggest that the meiotic chromosome axis stabilizes the loops.

### Similarities between mitotic metaphase and meiotic prophase chromosomes

The similarity of mitotic and meiotic chromosome *cis*-contact frequency (Fig 4I), as well as the conservation of mitotic and meiotic cohesin binding sites (Glynn *et al*, 2004), points at strong similarities between the folding of yeast chromosomes during meiotic prophase and mitotic metaphase. A unified perspective of the mitotic and meiotic programs has been shown for mammals, where the chromatin-loop array structure of meiotic mid-prophase chromosomes resembles their mitotic counterparts (Liang *et al*, 2015). In budding yeast, the chromatin loops that form during early anaphase of meiosis would therefore follow the same pattern than the ones that structure mitotic chromosomes during metaphase. Interestingly, we also noticed that in our experiments, non-adjacent Rec8 binding sites start to display preferential contacts at later time points (pachytene). The release of cohesin from the axis could result in the merging of loops into longer ones in a subset of the cells. Alternatively, a local condensation increase (Fig 4E) would bring the basis of the loops even closer, resulting in such a distant periodic signal. The contact patterns (clearly visible on the 3D representations of the maps; Fig 4D) show that chromosomes become thicker, as described for mammalian mitotic metaphase (Liang *et al*, 2015). What drives this compaction remains speculative, but given that the loops are already settled at early prophase, one could invoke restructuration of the axis meshwork and/or mechanical constraints that would bring their bases closer (Kleckner *et al*, 2004).

### Tracking homologs during meiotic prophase

The redesigned region now allows to distinguish the two homologs and track their 3D folding during meiotic prophase. The contacts between the two homologs display a ~3-fold increase, though no discrete contact pattern between them was observed. This suggests that chromatin loops on each homolog are not tightly associated or well aligned with each other. Alternatively, a cell-to-cell heterogeneous intermingling of pairs of homolog loops across the SC may also account for this blurred pattern. However, such contacts will be constrained by the size of the loops, with most loops having little freedom to reach one another across the ~200 nm made of the SC

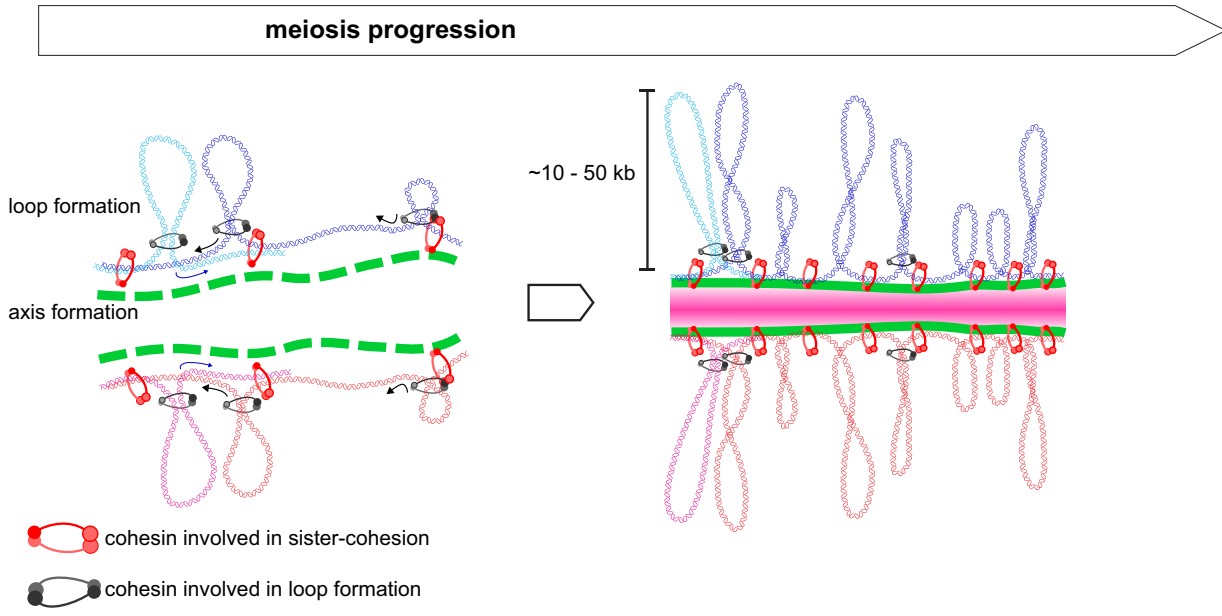

**Figure 7.    Proposed model for chromosome structuring during meiotic prophase.**

Replicating chromosomes undergo cohesin-mediated condensation possibly through loop extrusion mechanisms. As proposed for mitosis (Schalbetter *et al*, 2017), two sets of cohesins could act separately to mediate sister cohesion (red cohesin bridging the blue and black sisters) on the one hand, and loop formation on the other (other models can also be envisioned). For clarity, only one loop of each homolog sister is represented (magenta and pink loops). Extrusion of the DNA through cohesins is indicated by arrows. The axis stabilizes the loop boundaries, resulting in loops of various sizes. At zygotene and pachytene, the synaptonemal complex (pink) insulates the two homologs from each others.

and axis (Zickler & Kleckner, 2015). The nature of the non-specific contacts observed between the homologs remains therefore to be investigated to decipher to what extent the SC constitutes a barrier preventing direct contact between homologs. We showed that the polymorphisms introduced between the Syn-HiC and the native region along the homolog can also be used to investigate recombination events taking place within the region. Future studies will aim at deciphering the links between the higher-order organization of each homolog, their interplay, and the recombination events that take place within the entire region. We believe the Syn-HiC region and its unique features will constitute a powerful and convenient resource to do so.

**Refining the Syn-HiC design**

Overall, the design principles (including PCRTags; Materials and Methods) resulted in a total of 3,229 modified bases in a 144,558-bp window (~2%). In total, 743 codons were modified, but no change in the sequences of the corresponding proteins was introduced. A change in gene expression was observed for a subset of genes, probably as a result of modifications introduced in intergenic non-coding regions. The cause of these changes could be investigated in the future to improve design principles. Although we took great care in the design of the sequence and algorithm, this observation and our ongoing experiments nevertheless suggest this design is perfectible and could be optimized. Simplifying the design, notably through the removal of some restriction sites, could diminish the number of mutations introduced by a few hundreds and reduce the chance to modify gene expression or introduce deleterious mutations. Regarding the purpose of the

Syn-HiC region to investigate meiosis prophase, the synthetic design could also be adapted by removing polymorphisms around DSB hotspots of interest, to limit mismatch formation within recombination events that trigger the anti-recombinogenic activity of the mismatch repair machinery (Martini *et al*, 2011).

**Syn-HiC in other organisms**

The yeast genome presents a relatively homogeneous GC content and few repeated sequences. The gain in resolution achieved by redesigning restriction sites along a chromosome region should therefore be even higher in organisms with heterogeneous genomic content and will enable unbiased tracking of entire regions that are otherwise inaccessible to the Hi-C experiment. One could envision, for instance, assembling the redesigned chromosome in yeast (Benders *et al*, 2010), before replacing its native counterpart in the organisms of interest (such as a bacterium, or potentially mammalian cells; Koszul, 2016). Introducing SNPs through the reorganization of restriction sites could prove convenient to investigate the function and organization of repeated sequences in large, complex genomes and could be implemented in the Human Genome Project Write (Boeke *et al*, 2016). Finally, the restriction polymorphism introduced along the Syn-HiC design can also prove useful beyond Hi-C, taking advantage of the restriction assays developed in the past to track DNA recombination or repair events (Schwacha & Kleckner, 1994; Hunter & Kleckner, 2001; Piazza *et al*, 2018). This specific Hi-C-friendly design is, to our knowledge, the first in which where a large (144 kb) chromosome region is extensively redesigned and reassembled so that a genomic assay can now address chromosome metabolism questions otherwise difficult to tackle. It illustrates the

power of synthetic biology to boost, refine, and maybe reshape traditional molecular biology approaches.

# Materials and Methods

### Strains

The Syn-HiC region was assembled in the s288C and SK1 genetic backgrounds by transformation of BY4742 and ORT4602 (Sollier *et al*, 2004) to generate strains YRSG181 and YRSG189, respectively (see above). The SK1 diploid heterozygous for the Syn-HiC region (YRSG190) was obtained by crossing YRSG189 and ORT4601 (Sollier *et al*, 2004). Genotypes are provided in Table 2.

### Design principles of Syn-HiC chromosomes

We aimed at modifying the native sequence of a budding yeast chromosome according to our design principles while introducing as little modifications as possible. Because we were planning on re-assembling only a 150-kb window within the genome, we scanned through the overall sequence using a scoring quality function to look for the candidate regions qualifying as the ideal target, i.e., where our principles would introduce a minimal number of mutations. The starting material was the *S. cerevisiae* SK1 strain genome sequence and annotations available at the time (Liti *et al*, 2009) and a list of nine restriction enzymes (EcoRI, HindIII, NdeI, PstI, SacI, SacII, SalI, XbaI, XhoI, and DpnII). REs were selected based on their low cost and restriction efficiency. A genome index file was then computed that contained the following information for each base pair:

- Whether it consists of a "forbidden mutation" site, defined by us as follows: (i) start and stop codons of known ORFs; (ii) regulatory transcription pre-initiation complexes binding regions identified through ChIP-Seq exo, encompassing TATA-box binding sites (Rhee & Pugh, 2012); (iii) the consensus sequence of autonomously replicating sequences (ARS), i.e., the core sequence within *S. cerevisiae* replication origins (list of ARS obtained from oridb; Siow *et al*, 2012); (iv) intron borders; (v) centromeres; and (vi) tRNA.
- Whether the position belongs to a restriction site.
- Whether it belongs to an intergenic or coding region, and in the latter case, the codon it belongs to and its position.

Sliding windows of 150 kb moving with 10-kb steps were then generated over the entire genome.

In parallel, we defined the restriction pattern we wanted to generate:

- Regularly spaced intervals for 400, 1,500, 2,000 and 6,000 bp
- Gene promoter/terminator (substitutions within a coding sequence strongly preferred).

For each window, we computed all possible changes to apply to the genome so that all combinations of five out of the eight chosen 6-cutter enzymes were repositioned to generate all expected new restriction patterns. For each combination of five enzymes, all sites

were first removed from the genome before being reintroduced at ideal positions. A margin of error in the positioning of the "ideal" position was tolerated (10% of the window size) to maximize the probability of introducing only synonymous mutations within the coding sequence. Once a RS was positioned, the position of the adjacent RS was adjusted based on the newly positioned site so that overall, the distribution of RFs remains as close as possible to the theoretical distribution. Overall, for each enzyme, a quality score was computed for each window based on the difference between the expected distribution of the site and the real distribution. For each combination of enzyme, a global score corresponding to the sum of the individual scores of each enzyme was computed (see Fig EV1 for schema).

Overall, we selected the 10 "best" windows located at least at 150 kb from either a centromere or a telomere. The quality score was weighted by the presence of "forbidden positions" within the window, for instance, when a start codon overlaps a restriction site to be deleted. Finally, a manual curation, aiming at fixing potential conflicts (such as 2 RSs overlapping the same bases, or accidental re-creation of a RS of one enzyme when processing a second one), followed and was performed on the genome windows presenting the best quality scores.

We chose the final window based also on our research interests, i.e., containing at least two early replicating replication origins (Raghuraman *et al*, 2001; Siow *et al*, 2012), and several hotspots of meiotic DNA DSBs (Pan *et al*, 2011). We also attempted to avoid too many retrotransposable elements or other DNA repeats. The final window was positioned on chromosome IV::700,000–850,000, with restriction patterns as follows: DpnII ↔ 400-bp window; XbaI ↔ 1,500-bp window; HindIII ↔ 2,000-bp window; NdeI ↔ 6,000-bp window; and HhaI ↔ promoter/terminator (see summary on Fig EV2). A total of 1037 mutations were present in the sequence, the vast majority corresponding to the modifications necessary to reorganize DpnII RS (Table 1). Overall, 1,037 mutations were introduced, corresponding to 0.7% divergence.

Other modifications were introduced into the sequence. First, PCRTags similar to those used in the Sc2.0 design (Dymond *et al*, 2011; Annaluru *et al*, 2014) specific to either the native or synthetic sequence were also introduced within the window. Performing PCR using these primers allows testing for the presence and absence of the synthetic sequence and native sequence, respectively. PCRTags were manually curated to adapt them to the restriction design, and overall, 59 PCRTags out of 154 needed to be modified accordingly.

### Assembly of the redesigned chromosome

The redesigned sequence was split into 52 fragments of ~3,000 bp (i.e., blocks), with 200 bp overlaps between them. In addition, sequences corresponding to either of the auxotrophic marker genes *URA3* or *LEU2* were added to blocks 20, 37, and 52 (*URA3*) and blocks 11, 28, and 47 (*LEU2*), followed by 200-bp sequences of the wt neighboring chromosomal region. The replacement of the native sequence of strains BY4742 and SK1 with the redesigned blocks was performed through a succession of six transformations, up to 11 blocks at a time (Muller *et al*, 2012).

After each transformation, independent colonies were sampled and PCRs performed at the PCR tag positions to identify the transformants that have replaced all of the native sequence with the

redesigned one (Fig EV3). Upon the last transformation, the selected transformant genome was sequenced and the region 707,556–852,114 (144,558 bp; BY coordinates) was found to be replaced by the synthetic blocks.

### Growth rate analysis

Growth assays were performed to see whether the transformants exhibited changes in fitness. Little to no growth defect could be identified when blocks 1–47 replaced the native sequence in both BY and SK1 backgrounds. The final transformation with blocks 48–52 led repeatedly to the recovery of transformants exhibiting a slow-growth, petite phenotype (Slonimski, 1949), reflecting a block in the aerobic respiratory chain pathway and a decrease in ATP. However, crossing this petite strain with a wt strain gave respiratory-proficient diploids without growth defects.

### Synchronization of G1 cells

G1 cells of YRSG181 and BY4742 strains were recovered from exponential growing cultures through elutriation (Marbouty *et al*, 2014). The G1 daughter cells recovered through elutriation were suspended in fresh YPD at 30°C for 30 min, so they could recover from the elutriation procedure and their stay in PBS.

### Pre-growth and meiotic time course

Pre-growth and synchronous sporulation was adapted from Oh *et al* (2009). Briefly, cells patched on YPG-Agar plates (1% yeast extract, 2% peptone, 1.5% agar, 2% glycerol) from -80°C stocks were streaked on YPD plates (1% yeast extract, 2% peptone, 1.5% agar, 2% D-glucose, 0.004% adenine). A single colony was used to inoculate 5 ml YPD liquid culture and grown at 30°C up to saturation. The saturated culture was used to inoculate 350 ml of a freshly made (< 48 h) pre-sporulation medium (SPS; 0.5% yeast extract, 1% peptone, 1% potassium acetate, 1% ammonium sulfate, 0.5% potassium hydrogen phthalate, 0.17% yeast nitrogen base lacking all amino acids, two drops of anti-foaming agent) and grown with robust agitation (320 rpm) in 2.5-l baffled flasks at 30°C. The cells were washed with 200 ml and resuspended in 500 ml of pre-warmed sporulation media (SPM; 1% potassium acetate, 0.2× of uracil, arginine, and leucine, two drops of anti-foaming agent, and 7.5 µl 50% PEG350) and put back with robust agitation at 30°C. Samples were collected for Hi-C or Southern blot analysis (below) at 0, 3, and 4 h for the wild-type strain (YRSG190), and at 6 h for the wild-type (YRSG190) and *ndt80Δ*-arrested cells (strain YRSG154). Note that $t = 6$ h on the one hand and $t = 0$, 3, and 4 h on the other are from two different meiotic time courses. Since $t = 6$ h and pachytene-arrested cells gave very similar Hi-C patterns, we moved forward to compare these datasets with the ones obtained from the 0-, 3-, and 4-h kinetics.

### RNA isolation from yeast for RNA sequencing

Three independent RNA-seq libraries were generated for BY4742, SK1, and Syn-HiC strains. For each library, a single colony was grown in a 2 ml YPD culture overnight at 30°C. The next morning, 10 ml cultures in YPD were started from $10^6$ cells/ml until they

reached $2 \times 10^7$ cells/ml. The cells were pelleted by spinning at ~3,300 *g* at 4°C for 5 min. The pellet was resuspended in 0.5 ml of Tris–HCl (10 mM, pH 7.5) and transferred to a microfuge tube. The cells were pelleted again by spinning briefly and discarding the supernatant. The cells were resuspended in 400 µl RNA TES buffer (10 mM Tris–HCl pH 7.5, 10 mM EDTA, 0.5% SDS). RNA were treated with DNase TURBO (Invitrogen) and extracted twice with acid phenol/chloroform before precipitated and suspended in 50 µl of water.

### RNA-seq analysis of Syn-HiC

Single-end non-strand-specific RNA-seq of the YRSG181 (Syn-HiC) and its parental strain BY4742 was performed using Illumina NextSeq and standard TruSeq preparations kits, after depletion of ribosomal RNA. Reads were mapped using Bowtie2 (quality > 30) to the reference *S. cerevisiae* BY4742 and YRSG181 (Syn-HiC) genome. Gene differential expression was measured using DESeq2, with standard parameters.

### Hi-C experiments

Hi-C libraries were generated as described (Dekker *et al*, 2002; Lazar-Stefanita *et al*, 2017). Cells were cross-linked for 30 min with fresh formaldehyde (3% final concentration). To generate the libraries with different restriction enzymes, aliquots of $3 \times 10^9$ cells were resuspended in 10 ml sorbitol 1 M and incubated 30 min with DTT 5 mM and Zymolyase 100T ($C_{Final} = 1$ mg/ml) to digest the cell wall. Spheroplasts were washed with 5 ml of sorbitol 1 M, then with 5 ml of 1× restriction buffer (depending on the restriction enzyme used). Spheroplasts were then resuspended in 3.5 ml of the corresponding restriction buffer and split into three tubes (V = 500 µl) before a 20 min at 65°C incubated in SDS (3%). Cross-linked DNA was digested at 37°C overnight with the appropriate restriction enzyme (New England Biolabs; DpnII, HindIII, or NdeI). The digestion mix was then centrifuged and the pellets suspended and pooled into 400 µl of cold water. Depending on the sequence of the restriction site overhangs, extremities of the fragments were repaired in the presence of either 14-dCTP biotin or 14-dATP biotin (Invitrogen). Biotinylated DNA molecules were then incubated for 4 h at 16°C in the presence of 250 U of T4 DNA ligase (Thermo Scientific, 12.5 ml final volume). DNA purification was achieved through an overnight incubation at 65°C in the presence of 250 µg/ml proteinase K in 6.2 mM EDTA followed by the precipitation step in the presence of RNase.

The resulting Hi-C libraries were sheared and processed into Illumina libraries using custom-made versions of the Illumina PE adapters (Paired-End DNA sample Prep Kit; Illumina, PE-930-1001). Fragments of sizes between 400 and 800 bp were purified using a Pippin Prep apparatus (SAGE Science). Biotinylated molecules were purified using Dynabeads MyOne Streptavidin C1 beads, PCR-amplified, and paired-end-sequenced on an Illumina platform (HiSeq 2000; 2 × 75 bp).

### Processing of the reads and contact map generations

The raw data from each 3C experiment were processed as follows: First, PCR duplicates were collapsed using the six Ns

**Table 3.  Hi-C datasets used in this work, with corresponding accession numbers.**

| Figure panel | Name of the strain | Chr4 | Enzyme | Conditions | Reads (raw) | Accession number |
|---|---|---|---|---|---|---|
| 1D, 2A and C | YRSG181 | syn-HiC | DpnII | G1 elutriated | 22766486 | SRX4047313 |
| 2A and C | BY4741 | wt | DpnII | G1 elutriated | 14526516 | SRX4047314 |
| | BY4741 | wt | DpnII | G1 elutriated | 14831971 | SRX4047315 |
| 2B and D | YRSG181 | syn-HiC | HindIII | G1 elutriated | 25166666 | SRX4047316 |
| 2B and D | YKL053 | wt | HindIII | Cdc15-2 | 25065989 | SRX4047317 |
| 3A | YRSG181 | syn-HiC | DpnII | G1 elutriated (chr IV data)[a] | 22766486 | SRX4047313 |
| | YRSG181 | syn-HiC | DpnII | Early S (chr IV data)[a] | 13437325 | SRX2396527 |
| | YRSG181 | syn-HiC | DpnII | Mid S (chr IV data)[a] | 17194685 | SRX2396528 |
| | YRSG181 | syn-HiC | DpnII | Late S (chr IV data)[a] | 14565733 | SRX2396529 |
| | YRSG181 | syn-HiC | DpnII | G2/M arrest (chr IV data)[a] | 44733029 | SRX2396530 |
| | YRSG181 | syn-HiC | DpnII | G2/M released 20' (chr IV data)[a] | 31152533 | SRX2396531 |
| | YRSG181 | syn-HiC | DpnII | G2/M released 45' (chr IV data)[a] | 21213028 | SRX2396532 |
| | YRSG181 | syn-HiC | DpnII | Early S (chr IV data)[a] | 16002780 | SRX2396534 |
| | YRSG181 | syn-HiC | DpnII | Early S (chr IV data)[a] | 44575901 | SRX2396535 |
| | YRSG181 | syn-HiC | DpnII | Mid S (chr IV data)[a] | 49102855 | SRX2396536 |
| | YRSG181 | syn-HiC | DpnII | Mid S (chr IV data)[a] | 49916298 | SRX2396537 |
| | YRSG181 | syn-HiC | DpnII | G2/M (chr IV data)[a] | 25625208 | SRX2396538 |
| | YRSG181 | syn-HiC | DpnII | G2/M released 20' (chr IV data)[a] | 31990273 | SRX2396539 |
| | YRSG181 | syn-HiC | DpnII | G2/M released 45' (chr IV data)[a] | 34409100 | SRX2396540 |
| | YRSG181 | syn-HiC | DpnII | G2/M released 60' (chr IV data)[a] | 33540877 | SRX2396541 |
| | YRSG181 | syn-HiC | DpnII | G2/M released 90' (chr IV data)[a] | 35227228 | SRX2396542 |
| | YRSG181 | syn-HiC | DpnII | Asynchronous | 43649470 | SRX2396555 |
| | YRSG181 | syn-HiC | DpnII | G1 elutriation | 29657764 | SRX2396556 |
| | YRSG181 | syn-HiC | DpnII | G1 elutriation | 39064198 | SRX2396557 |
| 3B | YRSG181 | syn-HiC | DpnII | G1 elutriation + capture | 36279141 | SRX4047318 |
| | yLM539 | wt | DpnII | G1 elutriated (chr IV data)[b] | 35000000 | SRX1588814 |
| | yLM896 | wt | DpnII | G1 elutriated (chr IV data)[b] | 45200000 | SRX1588815 |
| | JDY451 | wt | DpnII | G1 elutriated (chr IV data)[b] | 40300000 | SRX1588817 |
| | YS031 | wt | DpnII | G1 elutriated (chr IV data)[b] | 29300000 | SRX1588804 |
| | JDY450 | wt | DpnII | G1 elutriated (chr IV data)[b] | 46500000 | SRX1588816 |
| | yXZX538 | wt | DpnII | G1 elutriated (chr IV data)[b] | 45100000 | SRX1588810 |
| | yXZX573 | wt | DpnII | G1 elutriated (chr IV data)[b] | 39100000 | SRX1588812 |
| | JDY448 | wt | DpnII | G1 elutriated (chr IV data)[b] | 32200000 | SRX1588806 |

**Table 3** (continued)

| Figure panel | Name of the strain | Chr4 | Enzyme | Conditions | Reads (raw) | Accession number |
|---|---|---|---|---|---|---|
| | JDY449 | wt | DpnII | G1 elutriated (chr IV data)[b] | 60800000 | SRX1588807 |
| | JDY446 | wt | DpnII | G1 elutriated (chr IV data)[b] | 26000000 | SRX1588805 |
| | JDY444 | wt | DpnII | G1 elutriated (chr IV data)[b] | 24600000 | SRX1588813 |
| | yYW0115 | wt | DpnII | G1 elutriated (chr IV data)[b] | 8900000 | SRX1588811 |
| 4, 5 | YRSG190 | syn/wt | DpnII | Meiosis, $t = 0$ h | 19040077 | SRX4047319 |
| | YRSG190 | syn/wt | DpnII | Meiosis, $t = 0$ h | 47334940 | SRX4047319 |
| | YRSG190 | syn/wt | DpnII | Meiosis, $t = 3$ h | 37056369 | SRX4047320 |
| | YRSG190 | syn/wt | DpnII | Meiosis, $t = 3$ h | 82303084 | SRX4047320 |
| | YRSG190 | syn/wt | DpnII | Meiosis, $t = 4$ h | 54612931 | SRX4047312 |
| | YRSG190 | syn/wt | DpnII | Meiosis, $t = 4$ h | 120645022 | SRX4047312 |
| | YRSG190 | syn/wt | DpnII | Meiosis, $t = 6$ h | 80684670 | SRX4213779 |
| | YRSG154 | syn/wt | DpnII | *ndt80Δ*-arrested, $t = 6$ h | 93426413 | SRX4213796 |
| | W303-1a | wt | HindIII | *Cdc20*-arrested[c] | 192027518 | SRR4292760 |
| RNA-seq | BY4741 | wt | n/a | | 90036018 | SRX4051932 |
| | BY4741 | wt | n/a | | 74934724 | SRX4051968 |
| | BY4741 | wt | n/a | | 88869023 | SRX4051974 |
| | YRSG190 | syn/wt | n/a | | 90994108 | SRX4051975 |
| | YRSG190 | syn/wt | n/a | | 87394100 | SRX4052000 |
| | YRSG190 | syn/wt | n/a | | 73318618 | SRX4052001 |

[a]From Lazar-Stefanita *et al* (2017).
[b]From Mercy *et al* (2017).
[c]From Schalbetter *et al* (2017).

present on each of the custom-made adapter and trimmed. Reads were aligned independently using Bowtie2 in its most sensitive mode against the *S. cerevisiae* reference genome (native genome) or against the *S. cerevisiae* reference adapted for the Syn-HiC region on chromosome IV (Syn-HiC genome). For SK1 strains, the new reference genome was recovered from Yue *et al* (2017). An iterative alignment procedure was used: For each read, the length of the sequence being aligned was gradually increased from 20 bp until the mapping became unambiguous (mapping quality > 30). Paired reads were aligned independently, and each mapped read was assigned to a restriction fragment. Re-ligation events have been filtered out using the information about the orientation of the sequences as described in Cournac *et al* (2012). Contact matrices were built for the wild-type regions (i.e., the entire region minus the chromosome IV Syn-HiC region) by binning the aligned reads into units of 5-kb bins. Maps were normalized using the sequential component procedure described in Cournac *et al* (2012).

DpnII and HindIII contact maps for the Syn-HiC region and its native counterpart were randomly resampled to present the same number of contacts. For statistical analysis of the Syn-HiC region, aligned reads were binned either into single restriction fragments (to illustrate the distribution of the reads along the synthetic region and its native counterpart presented in Fig 2A and B) or into units

(i.e., bins) of 600, 1,200, 2,400, 4,800 and 9,600 bp (Fig 2C and D). Contact maps were generated using the *levelplot* function of the R *lattice* package. Matrices for the synthetic region were subsequently obtained by extracting the diagonal blocks for bins falling in the 719,756–849,206-bp interval. Outliers have been removed from the matrices if the number of the contacts surpassed by 20 times the top 5‰ threshold of the number of contacts between restriction fragment pairs.

## Computation of the contact probability as a function of genomic distance

The decrease in contact probability $p(s)$ as a function of the genomic distance $s$ along chromosomes was computed for individual Hi-C experiments as follows: First, intra-chromosomal pairs of reads were partitioned by chromosome arms. Pairs oriented toward different directions or separated by less than 1.5 kb were discarded, as they may correspond to self-circularization events. For each chromosome, the remaining pairs were log-binned as a function of their genomic distance $s$ using the following formula: $bin = [\log 1.1(s)]$. The histogram was computed from the number of read pairs for each bin. This sum is weighed by the bin size $1.1^{(1+bin)}$, as well as the difference between the length of the chromosome and the genomic distance $s$.

### Capture-C

Following Hi-C library preparation, DNA was sheared in 150- to 200-bp fragments using a Covaris apparatus. DNA ends were repaired and adenylated according to the manufacturer's instructions using a SureSelect XT library kit (Agilent Technologies). As for other Hi-C Illumina libraries, custom paired-end adaptors were used and ligated to the ends. Biotinylated fragments were pulled down using Dynabeads MyOne Streptavidin C1 beads prior to pre-capture PCR amplification (six cycles). Biotinylated DNA matrix was separated from amplified DNA using a magnetic rack and stored for future usages. Capture of the genomic region of interest from the amplified DNA was performed according to the manufacturer's instructions using the custom-made SureSelect library corresponding to both the Syn-HiC and native chromosome IV regions. DNA was captured using Dynabeads MyOne Streptavidin T1 beads before the final post-capture PCR amplification using PE1 and PE2 primers for 16 cycles. Cleaned DNA was verified for size and quality on Bioanalyzer before paired-end (PE) sequencing on an Illumina platform (HiSeq 2000; 2 × 75 bp).

### Statistical analysis

The CV is defined as the ratio between the standard deviation and the mean of the contact histograms at fixed distance $s$; to take into account the finite-size effect, we discarded bins at the edge of the contact matrix in order to keep the statistics (number of bins) for different values of $s$ constant, up to $s < 15,000$ bp in DpnII and $s < 70,000$ bp in HindIII datasets. To show that the improvement is specific to the new restriction pattern and is unlikely to be found spontaneously within the genome, we compared the Syn-HiC results with seven regions of similar size along chromosome IV (460,856–590,306 bp; 590,306–719,756 bp; 849,206–978,656 bp; 978,656–1,108,106 bp; 1,108,106–1,237,556 bp; 1,2375,56–1,367,006 bp; 1,367,006–1,496,456 bp). The quality improvement was assessed by computing the logarithm of the ratio of the CVs of the Syn-HiC and native regions (Fig EV5).

### Southern blot analysis of crossing-over formation at the CCT6 hotspot

The *CCT6* locus was chosen because it is the strongest Spo11-induced DSB hotspot in the synthetic region (Pan *et al*, 2011). Cells were harvested and DNA was extracted as described (Oh *et al*, 2009), except that no cross-linking step was performed. The DNA was digested with NdeI and XbaI (New England Biolabs), migrated on a 1% UltraPure Agarose (Invitrogen) 1× TAE for 15 h at 70 V, and capillary transferred onto a Hybond-XL membrane (GE Healthcare) following the manufacturer's instructions. Southern blot hybridization was performed at 65°C in Church buffer (1% BSA, 0.25 M $Na_2HPO_4$ pH 7.3, 7% SDS, 1 mM EDTA) with a 1,104-bp-long radiolabeled probe corresponding to the rightmost region common to both the native and the Syn-HiC restriction fragments (obtained from SK1 genomic DNA with primers 5′-TGGTGAAG AACTCAGGATTC-3′ and 5′-CAGTTACAATGAAGTCCAGG-3′) and radiolabeled phage lambda DNA (molecular ladder). Radiolabeling was performed with $^{32}$P-αdCTP with the High Prime labeling kit (Roche) following the manufacturer's instructions. The membrane was washed and exposed overnight, and the storage Phosphor

Screen (GE Healthcare) was scanned on a Typhoon PhosphorImager (Molecular Dynamics). The length of the native and Syn-HiC parental fragments is 5,135 and 6,157 bp, respectively. CO formation generates two recombinants of 4,453 and 6,839 bp. Quantifications were performed with ImageJ 1.49v.

### Data availability

The datasets and computer code produced in this study are available in the following databases: Hi-C and RNA-seq data: Bioproject PRJNA464299 (https://www.ncbi.nlm.nih.gov/bioproject/?term = PRJNA464299).

The fasta sequences of the Syn-HiC region and its native counterpart are available as a supplementary file (Dataset EV1).

Previously published datasets used in this work are described in Table 3.

**Expanded View** for this article is available online.

### Acknowledgements

We thank Jef Boeke, Nancy Kleckner, and Gianni Liti for fruitful discussions and comments on the manuscript. We are also grateful to Jef Boeke for providing us with the Sc2.0 PCRTags sequences for chromosome 4. We thank Elodie Pirayre and Ivan Moszer for contributing to the initial steps of the design of the algorithm. We also thank Martial Marbouty, Axel Cournac, and Rémi Montagne for advices during the course of this work. Vittore Scolari and Héloïse Muller were recipients of Pasteur Roux Cantarini fellowships, and Aurèle Piazza was supported by the Framework Project 7 of the European Union (Marie Curie International Outgoing Fellowship 628355). This research was supported by funding to R.K. from the European Research Council under the 7th Framework Program (H2020 ERC grant agreement DLV-771813) and from ERASynBio and Agence Nationale pour la Recherche (IESY ANR-14-SYNB-0001-03), and to R.K., O.E., and B.L. from Agence Nationale pour la Recherche (MeioRec ANR-13-BSV6-0012).

### Author contributions

HM, SDD, BL, OE, NA, GF, and RK designed the sequence. HM assembled the Syn-HiC regions. HM and AP ran the meiotic time courses. AT did the Hi-C experiments with help from GM. VS analyzed the data and did the resolution analysis, with contributions from JM and LLS. AP analyzed meiotic recombination. RK conceived the study and wrote the manuscript, with contributions from JM, VS, HM, AP, and BL.

### Conflict of interest

The authors declare that they have no conflict of interest.

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
