## [Review Process File · Molecular Systems Biology]

Characterizing meiotic chromosomes' structure and pairing using a designer sequence optimized for Hi-C

Muller Héloïse, Scolari F. Vittore, Agier Nicolas, Aurèle Piazza, Mercy Guillaume, Lazar-Stefanita Luciana, Descorps-Declere Stephane, Espeli Olivier, Llorente Bertrand, Fischer Gilles, Mozziconacci Julien, and Koszul Romain

Review timeline:

First Submission:	17 th May 2016
Editorial Decision:	10 th June 2016
Second Submission:	27 th February 2018
Editorial Decision:	11 th April 2018
Revision received:	10 th June 2018
Editorial Decision:	14 th June 2018
Revision received:	18 th June 2018
Accepted:	20 th June 2018

Editor: Maria Polychronidou

Transaction Report:

1st Editorial Decision 10th June 2016

Thank you again for submitting your work to Molecular Systems Biology. We have now heard back from the three referees who agreed to evaluate your manuscript. As you will see below, the reviewers raise substantial concerns on your work, which unfortunately preclude its publication in Molecular Systems Biology.

Overall, the reviewers are not convinced that the proposed approach is likely to be broadly applied in analyses of chromatin conformation and they refer the lack of essential control experiments (i.e. showing that the sequence changes do not affect regulatory elements and gene expression levels). Moreover, they point out that no new biological conclusions are derived from the presented analyses. Considering the substantial concerns raised and the overall low level of support provided by the referees, we cannot consider the manuscript in its current form, as a methodology-focused short Report, for publication in Molecular Systems Biology.

Nevertheless, along the lines of the reviewers' comments, we acknowledge that the proposed approach addresses an important issue in Hi-C approaches. Therefore we would not be opposed to considering a new and extended manuscript (Article format) containing further analyses convincingly demonstrating that the approach can reveal new biological insights that cannot be derived with existing approaches. All other issues raised by the referees, including the important concerns raised on the potential disruptive effects of the sequence changes, would also need to be rigorously addressed.

REFeree REPORTS.

Reviewer #1:

The authors redesigned and reassembled a 145 kb region in yeast with regularly spaced restriction sites to investigate and potentially overcome biases in 3C based methods that result from the uneven distribution of restriction sites.

The authors show that they redesigned a region to contain evenly spaced restriction sites for DpnII, XbaI, HindIII and NdeI, at regularly spaced positions of 400 bp, 1500 bp, 2000 bp and 6000 bp, respectively. They performed Hi-C experiments in the strain carrying the synthetic region and in a wild type strain using DpnII and HindIII. They show that the read coverage and contact counts between binned fragments is more homogenous in the redesigned synthetic region compared to the same region in the wild type strain, leading to a higher resolution of the Hi-C experiment. They compared the coefficient of variation of the distributions of the number of contacts between bins as a measure of the signal to noise ratio, and show that the synthetic region has a higher signal to noise ratio compared to the wild type.

The authors conclude from these results that redesigning restriction fragments in the genome is a feasible way to increase the resolution of Hi-C and enable unbiased investigation of regions in the genome that could otherwise be difficult to visualise by 3C based methods due to a skewed restriction fragment distribution.

However, apart from the fact that redesigning and reassembling an entire region of interest is a lot of work, it also introduces changes in the original sequence that could potentially disrupt important regulatory elements. Though the authors have tried to only target synonymous mutations in coding regions, and avoid forbidden mutations (in ARS, telomere regions, retrotransposable elements, mating type loci, tRNA, Sn/Sno RNA, rDNA, ncRNA, intron motives, TATA box) 'when possible', the chances that (unknown) important regulatory elements will be disrupted make this approach not incredibly useful for studying the organisation of chromatin.

It is interesting to see how redistributing the restriction fragments addresses an important bias in 3C resulting from the uneven distribution of restriction fragments. However, comparisons between restriction enzymes that cut chromatin more or less frequently (e.g. 4-cutters like DpnII and 6-cutters like HindIII) have already shown that using a more frequently cutting enzyme (that cuts the chromatin more evenly) increases the coverage and resolution of 3C-based experiments.

Reviewer #2:

Comments to authors

The article entitled "Redesigning chromosomes to optimize conformation capture (Hi-C) assays" by Heloise and co-workers introduces a nice trick to assess the effect on Hi-C experiments of the distribution of restriction sites in the genome. This is a point that has been thoroughly analyzed by many authors before using theoretical approaches aiming at normalizing 3C-based datasets. However, this is the first time that a genome (yeast) is modified to contain a relatively long region with equally spaced restriction sites with the goal of assess its effect in 3C-based experiments. Next, the authors have performed Hi-C on the yeast strain with the synthetic region and shown that such region contains more homogenous distribution of interactions that are likely to better reflect true interactions than regions of the genome with uneven distribution of restriction sites. Overall, the idea is innovative, the implementation correct and the analysis well performed. The conclusions of the work, although not biologically relevant, are likely to provide a framework for better designing Hi-C experiments. There are no major issues with the work that need to be addressed. In fact, this reviewer would have liked to see a more detailed analysis of the results, which is now limited by the "Report" format chosen.

Minor comments:

- Figure 1C. Both panels miss the labels/units in the y axis.
- Figure 1A/B. Could the authors make a clearer plots. As for now, they do not really convey the message. The vertical dark lines obscure the actual data.
- Figure 1C/D. Middle plots do not have axis labels.

- Page 3. "Fig D" should be "Fig 1D"
- There is a typo in line 4 of second paragraph of results. "designer" should be "designed".

Reviewer #3:

Muller et al. engineer a 150kb segment of the yeast genome with evenly placed restriction enzyme cut sites, and perform Hi-C on this modified genome. They observe a modest improvement in uniformity of contact frequencies for frequent cutter DpnII, and a significant improvement for infrequent-cutter HindIII. They propose this as a tool for future studies of genome architecture. This study addresses an important technical issue in Hi-C approaches, namely the uneven location of cut sites, and as such is of value. However, while this approach is interesting, we have concerns about its usefulness, and we would like to see more control experiments to verify that the approach does not alter the local chromatin landscape and can detect features reproducibly demonstrated in traditional Hi-C studies.

We have several major concerns with this study:

1. Muller and colleagues do not provide enough evidence that their introduction of ~3000 mutated base pairs in a 150kb span do not disrupt the chromatin landscape of that region. The only assay used is growth rate which is a rather coarse measure of functionality. The observation of no striking effect on growth rates does not necessarily imply that the chromatin landscape within this 150kb window is not significantly altered. At a minimum, the authors should perform RT-PCR to known genes in this region to show that their transcription levels have not changed. Ideally, they would provide other evidence that chromatin state and shape had not significantly changed upon the introduction of 3000 mutated base pairs such as analysis of local chromatin structure by biochemical or imaging.
2. They do not specify whether reproducible structural features of chromatin observed in Hi-C are reproduced in their redesigned chromosome. Are there any loops or TAD boundaries within this 150kb region, and if so are they observed in the redesigned chromosome as well as the wild type chromosome? The current analysis only looks at contact frequencies but not actual structural features of chromatin.
3. They propose this as a demonstration of the application of genome engineering to Hi-C, suggesting perhaps that redesigning whole genomes be added as a normal portion of the Hi-C workflow in order to improve data quality. This does not seem, to us, a realistic suggestion. If this is not their intended claim, they should specify exactly how genome engineering will shed light on Hi-C data.

Furthermore, we have several minor critiques:

1. In the first paragraph, it is mentioned that coding sequences were preferentially targeted for mutation, but no list of genes within the 150kb mutated region is included. We would like a list of genes within the 150kb region added to the supplement.
2. In the second paragraph, the authors say "Interestingly, careful examination of this distribution indicates that besides its own length, the capture frequency of a given fragment is also influenced by the length of its neighbors." But they do not specify either where, in the data, this is shown or why it is interesting or novel.
3. In all of their heat maps, the color-scales are variable; and they use this to illustrate their claim that heterogeneity in the data is consistently higher in WT as compared to their redesigned strain. Heat maps will be easier to compare if color-scales are equalized, and another metric or separate panel can demonstrate data heterogeneity.
4. They claim that their genome-engineering approach is preferable for tracking trans interactions, but do not mention whether they observe trans interactions in their data set, nor do they include data about detection of trans interactions.

In all, we think that this technology is clever, but we feel that some control experiments are missing. In several places, the authors make smaller claims that they do not back up with data. And we do not think their suggestion that this approach could be used routinely within the normal Hi-C workflow is reasonable at this time.

Point by Point response.

Reviewer #1:

The authors redesigned and reassembled a 145 kb region in yeast with regularly spaced restriction sites to investigate and potentially overcome biases in 3C based methods that result from the uneven distribution of restriction sites.

The authors show that they redesigned a region to contain evenly spaced restriction sites for DpnII, XbaI, HindIII and NdeI, at regularly spaced positions of 400 bp, 1500 bp, 2000 bp and 6000 bp, respectively. They performed Hi-C experiments in the strain carrying the synthetic region and in a wild type strain using DpnII and HindIII. They show that the read coverage and contact counts between binned fragments is more homogenous in the redesigned synthetic region compared to the same region in the wild type strain, leading to a higher resolution of the Hi-C experiment. They compared the coefficient of variation of the distributions of the number of contacts between bins as a measure of the signal to noise ratio, and show that the synthetic region has a higher signal to noise ratio compared to the wild type.

The authors conclude from these results that redesigning restriction fragments in the genome is a feasible way to increase the resolution of Hi-C and enable unbiased investigation of regions in the genome that could otherwise be difficult to visualise by 3C based methods due to a skewed restriction fragment distribution.

However, apart from the fact that redesigning and reassembling an entire region of interest is a lot of work, it also introduces changes in the original sequence that could potentially disrupt important regulatory elements. Though the authors have tried to only target synonymous mutations in coding regions, and avoid forbidden mutations (in ARS, telomere regions, retrotransposable elements, mating type loci, tRNA, Sn/Sno RNA, rDNA, ncRNA, intron motives, TATA box) 'when possible', the chances that (unknown) important regulatory elements will be disrupted make this approach not incredibly useful for studying the organisation of chromatin.

It is interesting to see how redistributing the restriction fragments addresses an important bias in 3C resulting from the uneven distribution of restriction fragments. However, comparisons between restriction enzymes that cut chromatin more or less frequently (e.g. 4-cutters like DpnII and 6-cutters like HindIII) have already shown that using a more frequently cutting enzyme (that cuts the chromatin more evenly) increases the coverage and resolution of 3C-based experiments.

We agree that we introduced many changes in the sequences that may not be, ultimately, useful, and there is room for simplification if one wants to exploit this approach in other organisms. However, the genetics robustness of yeast allowed it to accept quite a large amount of mutations without major impact on its growth rate or on other metabolic processes monitored in this revised manuscript. In addition, the field of large chromosome assembly is becoming increasingly popular, and this design or equivalent are likely to be implemented by other groups, even though

their questions are not primarily focused on genome folding (“just in case” strategy). We had several requests about the design from groups working in bacteria or worm, for instance.

In this revised version, we show that the syn-3C chromosome not only allows us to monitored with more accuracy the chromosomal region, leading us to provide a rigorous definition of a Hi-C experiment resolution, but also allow us to track homologs in a diploid strain undergoing meiosis.

We provide a first insight on the conventional meiotic program using Hi-C, and show that our design provides a very powerful tool to track homologous recombination in the context of higher order chromosomal architecture.

Reviewer #2:

Comments to authors

The article entitled "Redesigning chromosomes to optimize conformation capture (Hi-C) assays" by Heloise and co-workers introduces a nice trick to assess the effect on Hi-C experiments of the distribution of restriction sites in the genome. This is a point that has been thoroughly analyzed by many authors before using theoretical approaches aiming at normalizing 3C-based datasets. However, this is the first time that a genome (yeast) is modified to contain a relatively long region with equally spaced restriction sites with the goal of assess its effect in 3C-based experiments. Next, the authors have performed Hi-C on the yeast strain with the synthetic region and shown that such region contains more homogenous distribution of interactions that are likely to better reflect true interactions than regions of the genome with uneven distribution of restriction sites. Overall, the idea is innovative, the implementation correct and the analysis well performed. The conclusions of the work, although not biologically relevant, are likely to provide a framework for better designing Hi-C experiments. There are no major issues with the work that need to be addressed. In fact, this reviewer would have liked to see a more detailed analysis of the results, which is now limited by the "Report" format chosen.

Minor comments:

- Figure 1C. Both panels miss the labels/units in the y axis.
- Figure 1A/B. Could the authors make a clearer plots. As for now, they do not really convey the message. The vertical dark lines obscure the actual data.
- Figure 1C/D. Middle plots do not have axis labels.
- Page 3. "Fig D" should be "Fig 1D"
- There is a typo in line 4 of second paragraph of results. "designer" should be "designed".

We thank the referee for these comments. We hope to have addressed the minor issues raised in the deeply revised work we are now submitting as a full article. We now have completed the work with a more detailed statistical analysis of the results, and included a “biological” experiment that we hope will be of interest to the referee.

Reviewer #3:

Muller et al. engineer a 150kb segment of the yeast genome with evenly placed

restriction enzyme cut sites, and perform Hi-C on this modified genome. They observe a modest improvement in uniformity of contact frequencies for frequent cutter DpnII, and a significant improvement for infrequent-cutter HindIII. They propose this as a tool for future studies of genome architecture.

This study addresses an important technical issue in Hi-C approaches, namely the uneven location of cut sites, and as such is of value. However, while this approach is interesting, we have concerns about its usefulness, and we would like to see more control experiments to verify that the approach does not alter the local chromatin landscape and can detect features reproducibly demonstrated in traditional Hi-C studies.

We have several major concerns with this study:

1. Muller and colleagues do not provide enough evidence that their introduction of ~3000 mutated base pairs in a 150kb span do not disrupt the chromatin landscape of that region. The only assay used is growth rate which is a rather coarse measure of functionality. The observation of no striking effect on growth rates does not necessarily imply that the chromatin landscape within this 150kb window is not significantly altered. At a minimum, the authors should perform RT-PCR to known genes in this region to show that their transcription levels have not changed. Ideally, they would provide other evidence that chromatin state and shape had not significantly changed upon the introduction of 3000 mutated base pairs such as analysis of local chromatin structure by biochemical or imaging.

We have generated a series of RNA-seq experiments that show that most genes are not affected by the design, even though 6 out of ~70 genes appear slightly deregulated. We also have performed meiosis synchronization using a diploid strain, and verify the redesigned region behaves as wild-type ones. We believe that overall these controls and experiments answer the referee's legitimate concerns.

2. They do not specify whether reproducible structural features of chromatin observed in Hi-C are reproduced in their redesigned chromosome. Are there any loops or TAD boundaries within this 150kb region, and if so are they observed in the redesigned chromosome as well as the wild type chromosome? The current analysis only looks at contact frequencies but not actual structural features of chromatin.

Unlike several other eukaryotes, budding yeast chromosome do not display TADs or higher order structural features readily apparent in wild-type Hi-C maps (see Duan et al., 2010 and our recent work Mercy 2017 and Lazar-Stefanita 2017). This may be due to the compactness of the genome, and/or the absence of enhancer sequences, etc. Therefore, we could not analyze such features in the redesigned region.

However, we have now included Hi-C experiments performed during meiotic prophase, a condition where domains and loops are being generated along the chromosome. We believe this is the first time that meiotic loops are being tracked genomewide, and a formal demonstration of their existence in yeast provided. We show that we can distinguish both homologs at the redesigned region, paving the way to future in-depth study of the functional interplay between the higher order folding of the chromosome and the meiotic recombination events that take place at this stage.

3. They propose this as a demonstration of the application of genome engineering to Hi-C, suggesting perhaps that redesigning whole genomes be added as a normal

portion of the Hi-C workflow in order to improve data quality. This does not seem, to us, a realistic suggestion. If this is not their intended claim, they should specify exactly how genome engineering will shed light on Hi-C data.

Indeed, this was not what we meant and it would be indeed unrealistic to redesign chromosomes for each type of Hi-C experiment. However, there are now several projects aiming at redesigning large chromosomal regions of a variety of species. The genome project-write (GP-write; Boeke et al., Science 2016) initiative for instance, or also a number of smaller projects aiming at redesigning parts of bacterial chromosomes (*E. coli*, *B. subtilis*, etc.) The design we present here has received a lot of interest from several of the groups involved in these projects. Not necessarily the complete design, but for instance the idea to insert a HindIII site every kb, which could already alleviate biases and limitations over large regions.

Furthermore, we have several minor critiques:

1. In the first paragraph, it is mentioned that coding sequences were preferentially targeted for mutation, but no list of genes within the 150kb mutated region is included. We would like a list of genes within the 150kb region added to the supplement.

We now indicate the window of the genes (ORF YDR127w up to YDR196c) included within the Syn-HiC region.

2. In the second paragraph, the authors say "Interestingly, careful examination of this distribution indicates that besides its own length, the capture frequency of a given fragment is also influenced by the length of its neighbors." But they do not specify either where, in the data, this is shown or why it is interesting or novel.

We have removed the sentence but this trend appears now clearly on the modified Figure 2.

3. In all of their heat maps, the color-scales are variable; and they use this to illustrate their claim that heterogeneity in the data is consistently higher in WT as compared to their redesigned strain. Heat maps will be easier to compare if color-scales are equalized, and another metric or separate panel can demonstrate data heterogeneity.

We have equalized, when possible, the colorscales. But these come sometimes from very different experiments (Capture C, meiosis, homolog/heterolog) and a complete homogenization was not always possible. We hope the referee will find the figures clear nevertheless.

4. They claim that their genome-engineering approach is preferable for tracking trans interactions, but do not mention whether they observe trans interactions in their data set, nor do they include data about detection of trans interactions.

We now discuss trans interactions between homologs during meiosis prophase.

In all, we think that this technology is clever, but we feel that some control experiments are missing. In several places, the authors make smaller claims that they do not back up with data. And we do not think their suggestion that this approach could be used routinely within the normal Hi-C workflow is reasonable at this time.

Thank you again for submitting your work to Molecular Systems Biology. We have now heard back from the two referees who agreed to evaluate your study. Reviewer #1 is the previous reviewer #3 who evaluated the earlier version of your work (MSB-16-7068). Reviewer #2 is a new reviewer, who was asked to evaluate the study afresh. As you will see below, the reviewers appreciate that the proposed approach sounds potentially useful. They raise however a series of concerns, which we would ask you to address in a revision of the manuscript.

Notably, reviewer #2 is concerned that as it stands the level of biological insight remains rather limited. Along the lines of this reviewer's comments we think that adding some further analyses performed in a perturbation context (e.g. in a *rec8* mutant as the reviewer suggests) would significantly enhance the overall impact of the study.

If you feel you can satisfactorily deal with these points listed by the referees, you may wish to submit a revised version of your manuscript. Please attach a covering letter giving details of the way in which you have handled each of the points raised by the referees. A revised manuscript will be once again subject to review and you probably understand that we can give you no guarantee at this stage that the eventual outcome will be favorable.

REFEREE REPORTS.

Reviewer #1:

The authors have addressed our major point of assessing the degree of functional change introduced by the engineering steps. They now show that 10% of genes in the target region are affected. While this still raises issues as to how practically applicable this approach is when the goal is to test for function or to relate structure to function, the transparent disclosure of these effects in the manuscript will allow others to assess this issue.

The authors have addressed all our other concerns.

Reviewer #2:

In their manuscript, the authors describe chromosome design in budding yeast and the application of Hi-C for the study of chromosome organization in meiosis. The manuscript consists of two parts that are to some extent independent. In the first part of the manuscript the authors discuss the design of a 145kb region such that restriction sites of 4 different enzymes are regularly spaced. They demonstrate this results in higher quality Hi-C data due to the tighter distribution of restriction fragment size. In the second part of the manuscript, the authors perform standard Hi-C for 3 timepoints in meiosis. Using wildtype yeast strains, they observe loss of centromeric interchromosomal contacts, chromosome condensation, and evidence of *rec8*-mediated structure (possibly loops). Finally, they use a strain which is heterozygote for their designed chromosome to study the contacts of homologs and observe increased interaction between homologs.

Overall the study is designed well and the manuscript is clearly written. The technology for chromosome design is impressive but not new, but the concept of using this to improve Hi-C data is novel. I also found the in-depth analysis of Hi-C resolution based on restriction fragment size distribution to be novel and interesting. However, the advance in this first part of the paper is more of a technical nature in of interest mainly for labs performing Hi-C. The biological findings in the second part of the paper are interesting and novel, though they mainly recapitulate observations made by other techniques. Still, this type of genomic data is much more powerful than microscopy and will be useful for further study by others. It is slightly disappointing that the only connection between the two parts of the paper turns out not to reveal much in terms of new biological findings (the use of the designed chromosome for observing a general increase in homolog interaction).

I recommend accepting the paper. However, I believe that adding Hi-C measurements of a perturbation (e.g. *rec8* mutant) could significantly increase the impact of the biological findings by

starting to establish mechanisms.

Minor comments:

1. The interhomolog Hi-C maps (Figure 5c) look surprisingly featureless, as I expected to observe some homolog juxtaposition. The authors mention a weak diagonal, but I cannot see it. I suggest adding a more thorough analysis of these maps.
2. While it indeed seems that there is some domain-like structure associated with rec8 binding, I would be more cautious in interpreting these as rec8 loops, given the correlative nature of this observation and the limitations of Hi-C in observing a population average. For example, the phrase "direct evidence of the establishment of cohesin-bound meiotic loops" is a bit too strong in my opinion.
3. In my opinion, 3D models of Hi-C maps mostly represent a way of visualizing the Hi-C data. It is not surprising that what is observed in the maps will also be found in the 3D model. Thus I would change "strikingly recapitulated by the 3D representation of the 2D maps".
4. It is not clear to me why the designed fragment length distribution is normal. Shouldn't they exactly reflect the design? Is this due to imperfect digestion?
5. It is not clear if matrix normalization/correction was used, and if so in which matrices.
6. I did not see any experimental details on the synchronization in the Methods section.
7. In the discussion of restriction fragment size, I suggest mentioning Hi-C strategies based on MNase and DNase.
8. In the introduction, the inability to differentiate between isogenic chromosomes is a limitation of all genomic techniques, not just Hi-C.
9. It would be helpful to mention the six mis-regulated non-dubious ORFs by gene name.
10. Some language editing is required.

1st Revision - authors' response

10th June 2018

Point-by-Point response.

Reviewer #1:

The authors have addressed our major point of assessing the degree of functional change introduced by the engineering steps. They now show that 10% of genes in the target region are affected. While this still raises issues as to how practically applicable this approach is when the goal is to test for function or to relate structure to function, the transparent disclosure of these effects in the manuscript will allow others to assess this issue.

The authors have addressed all our other concerns.

We thank the referee for his comments and suggesting to add the RNA seq analysis.

Reviewer #2:

In their manuscript, the authors describe chromosome design in budding yeast and the application of Hi-C for the study of chromosome organization in meiosis. The manuscript consists of two parts that are to some extent independent. In the first part of the manuscript the authors discuss the design of a 145kb region such

that restriction sites of 4 different enzymes are regularly spaced. They demonstrate this results in higher quality Hi-C data due to the tighter distribution of restriction fragment size. In the second part of the manuscript, the authors perform standard Hi-C for 3 timepoints in meiosis. Using wildtype yeast strains, they observe loss of centromeric interchromosomal contacts, chromosome condensation, and evidence of *rec8*-mediated structure (possibly loops). Finally, they use a strain which is heterozygote for their designed chromosome to study the contacts of homologs and observe increased interaction between homologs.

Overall the study is designed well and the manuscript is clearly written. The technology for chromosome design is impressive but not new, but the concept of using this to improve Hi-C data is novel. I also found the in-depth analysis of Hi-C resolution based on restriction fragment size distribution to be novel and interesting. However, the advance in this first part of the paper is more of a technical nature in of interest mainly for labs performing Hi-C. The biological findings in the second part of the paper are interesting and novel, though they mainly recapitulate observations made by other techniques. Still, this type of genomic data is much more powerful than microscopy and will be useful for further study by others. It is slightly disappointing that the only connection between the two parts of the paper turns out not to reveal much in terms of new biological findings (the use of the designed chromosome for observing a general increase in homolog interaction).

I recommend accepting the paper. However, I believe that adding Hi-C measurements of a perturbation (e.g. *rec8* mutant) could significantly increase the impact of the biological findings by starting to establish mechanisms.

We thank the referee for the careful reading of the manuscript and the comments. Indeed, the first part of the paper aims at describing the Syn-HiC design and providing new statistics and insights on Hi-C data, while the purpose of the second part objective was show that such an approach is useful to address biological question(s). With this in mind, we performed a proof-of-concept study showing that the syn-HiC chromosome allows tracking the pairing and structure of both homologs in an isogenic background, while allowing us to "capture" crossing-over

formation along the region (we showed that for one meiotic hotspot, but the quantification can be extended to multiple neighboring positions).

This analysis paves the way to a future in-depth investigation of the interplay between higher-order architecture of chromosomes and meiotic recombination, something that has never been addressed before due to the limitations that are now overcome. The referee is therefore right to say that it would be interesting to test mutants using the approach to dissect more of the mechanistic interplays between the structures, and molecular partners. However, we consider this is a bit premature at this stage. The *rec8Δ* mutant is definitely highly relevant and on our to-do list. We started investigating other mutants such as *mek1Δ*, but we figured out that adding these data lead to considerations (such as calling for other mutants, etc.) that go way beyond what can be reasonably included in this paper which is already massive. Therefore, focusing on the description of the wild-type analysis is already an important step forward that will be of interest to the meiosis community. However, we have strengthened the present analysis by adding a later time point to the analysis (6h into SPM) as well as pachytene arrested cells using a *ndt80Δ* mutation. The later experiment is important in the sense that in these conditions cells are robustly blocked at pachytene, a stage where homologs are synapsed, and chromatin organized as arrays of loops along chromosomes axis. *ndt80Δ*-arrested cells contact maps are very similar to those recovered for the t=6h time point. This similarity supports that the gradual changes observed in Hi-C maps from time points 0, 3, 4 to 6 h of the meiotic time course do reflect expected structural changes in nuclear organization that accompany premeiosis to pachytene (Figure 4). By reprocessing the data at 2.5 kb resolution, and increasing the number of useful reads, we were also able to visually identify loops-like signal involving Rec8 binding sites (Figure 5). More experiments will further dissect the interplay between these structural features and homologous recombination, and we now insist that this proof of concept study paves the way to exciting molecular work that we describe in a new discussion section.

Minor comments:

1. The interhomolog Hi-C maps (Figure 5c) look surprisingly featureless, as I expected to observe some homolog juxtaposition. The authors mention a weak diagonal, but I cannot see it. I suggest adding a more thorough analysis of these maps.

That homolog DNA does not juxtaposed closely is indeed, in our eyes, a surprising result. We discuss this, and propose that although chromosome axes are bridged by the SC, chromatin loops sequences do not appear closely juxtaposed. Obviously homologs should locally touch each other at recombination sites but whether our technique is able to capture these events remain to be determined. We added a p(s) analysis of these contacts (Figure 6C), showing they gradually increase over time, but remain always relatively low (never close to the tight diagonal that characterizes intrachromosomal contacts). More experiments will be needed to investigate the strength and specificity of contacts, for instance by inducing a controlled meiotic DSB site, and following a recombination event between the homologs.

2. While it indeed seems that there is some domain-like structure associated with rec8 binding, I would be more cautious in interpreting these as rec8 loops, given the correlative nature of this observation and the limitations of Hi-C in observing a population average. For example, the phrase "direct evidence of the establishment of cohesin-bound meiotic loops" is a bit too strong in my opinion.

We now describe (and discuss) with greater details (and references) the works that led to the characterization of loops in the past. One can note that these works are of two natures. On the one hand, imaging data showing loop arrays along pachytene chromosomes. On the other hand, the deposition of axis and axis-associated proteins (Rec8, Red1, etc.) at discrete positions along the DNA characterized by ChIP experiments performed over populations. The current vision that ~20 kb loops are bound to the axis by these molecular complexes is therefore, we believe, correlative and population-based.

Our data show clearly that chromosomes become organized as loop-like structures, centered on rec8 binding sites, exactly how it was suggested by previous data. These loops are of different sizes, which closely match the inter-Rec8 enrichment sites observed by ChIP (Ito et al. 2014) and are probably quite similar to the loops proposed to structure mitotic metaphase chromosomes since mitotic and meiotic

cohesins are deposited at overlapping binding sites. Also, a side by side comparison of p(s) of *cdc20* mitotic arrested cells and meiotic pachytene chromosomes show the contact frequencies at these two stages are very similar, suggesting, again, a very similar structure of chromosomes at both stages.

Therefore we think that, even though they remain correlative, our results go one step further in the demonstration of the existence of meiotic loops than the previous coincident cytology and CHIP data, which were already held by the community as strong evidence of their existence. We nevertheless agree that in absence of a mutant the claim may be a bit too strong and we have pondered the sentences.

3. In my opinion, 3D models of Hi-C maps mostly represent a way of visualizing the Hi-C data. It is not surprising that what is observed in the maps will also be found in the 3D model. Thus I would change "strikingly recapitulated by the 3D representation of the 2D maps".

Indeed, as these are not models but another way to represent the HiC data, we rephrase this sentence into the following:

“Individual 2D maps can also be represented as 3D structures to illustrate the transformation of chromosomes into well-individualized entities throughout prophase (Fig. 4D; Lesne et al, 2014).”

4. It is not clear to me why the designed fragment length distribution is normal. Shouldn't they exactly reflect the design? Is this due to imperfect digestion?

The sentence describing the distribution of redesigned restriction fragments in panel 1C as normal was clearly a mistake, this distribution is not normal. The constraints we impose led to the present distribution where most segments are of the size we aimed at, or larger, and few are smaller. We changed the sentence accordingly.

5. It is not clear if matrix normalization/correction was used, and if so in which matrices.

Thanks for pointing this out! We now indicate in the figure legends precisely for the different maps or analysis whether the maps were normalized, or not (the color scale is also an indication).

6. I did not see any experimental details on the synchronization in the Methods section.

The meiotic synchronization was described in the chapter “*Pre-growth and sporulation of yeast*”. We now added a chapter “*Synchronization of G1 cells*”, which was missing, in the Methods.

Synchronization of G1 cells

G1 cells of YRSG181 and BY4742 strains were recovered from exponential growing cultures through elutriation (Marbouty *et al*, 2014). The G1 daughter cells recovered through elutriation were suspended in fresh YPD at 30°C for 30 min, so they could recover from the elutriation procedure (i.e., stay in PBS).

We also added how the $p(s)$ from panels 4H, I and 6B are computed.

Computation of the contact probability as a function of genomic distance.

The decrease in contact probability $p(s)$ as a function of the genomic distance s along chromosomes was computed for individual Hi-C experiments as follow: first, intra-chromosomal pairs of reads were partitioned by chromosome arms. Pairs oriented towards different directions or separated by less than 1.5 kb were discarded, as they may correspond to self-circularization events. For each chromosome, the remaining pairs were log-binned as a function of their genomic distance s using the following formula: $bin = \lceil \log_{1.1}(s) \rceil$. The histogram computed from the number of read pairs for each bin. This sum is weighed by the bin-size $1.1^{(1+bin)}$, as well as the difference between the length of the chromosome and the genomic distance s .

7. In the discussion of restriction fragment size, I suggest mentioning Hi-C strategies based on MNase and DNase.

We have added the following chapter in the introduction.

“Approaches using modified restriction patterns have been used to increase/improve the resolution of 3C-based approaches, such as DNase Hi-C and Micro-C (Hsieh *et al*, 2015; Ramani *et al*, 2016). DNase-HiC captures contacts between open chromatin sites sensitive to DNase. These sites are found approximately every 3 kb along the yeast genome (Ma *et al*, 2015), and therefore the resolution reachable through DNase Hi-C remains limited. Micro-C, on the other hand, exploits micro-Coccal nuclease (Mnase) to digest DNA rather than a restriction enzyme (Hsieh *et al*, 2015, 2016). This approach generates nonspecific

cuts in-between nucleosomes (every ~160 bp), resulting in a relatively regular restriction pattern, with heterogeneities resulting from the distribution of nucleosome free regions. Short-range chromosome contacts captured by Micro-C identified large chromosomal domains within the yeast genome separated by highly expressed genes. While providing a high resolution of intrachromosomal contacts, this approach nevertheless suffers from the same limitations of classical Hi-C when it comes to the inability to track homologs and repeated regions.”

8. In the introduction, the inability to differentiate between isogenic chromosomes is a limitation of all genomic techniques, not just Hi-C.

We now generalize this limitation in the introduction:

“The second limitation is common to all genomic approaches, and reflects the fact that identical sequences cannot be tracked because the sequencing reads cannot be mapped unambiguously along the genome, abolishing the possibility to track homologous chromosomes in isogenic backgrounds.”

9. It would be helpful to mention the six mis-regulated non-dubious ORFs by gene name.

We have included a table listing these genes in Figure EV3

10. Some language editing is required.

We proofread the article with the help of a native English speaker.

2nd Editorial Decision

14th June 2018

Thank you again for sending us your revised manuscript. We are now satisfied with the modifications made and we think that the study is suitable for publication in Molecular Systems Biology.

Before we formally accept the study for publication we would ask you to address some remaining editorial issues listed below.

Corresponding Author Name: Romain Koszul

Manuscript Number: MSB-18-8293